# HiTIC-Monthly: A Monthly High Spatial Resolution (1 km) Human Thermal Index Collection over China during 2003–2020

Hui Zhang[1], Ming Luo[1,3*], Yongquan Zhao[2*], Lijie Lin[4], Erjia Ge[5], Yuanjian Yang[6], Guicai Ning[3], Jing Cong[7], Zhaoliang Zeng[8], Ke Gui[9], Jing Li[10], Ting On Chan[1], Xiang Li[1], Sijia Wu[1], Peng Wang[1], Xiaoyu Wang[1]

[1]School of Geography and Planning, and Guangdong Key Laboratory for Urbanization and Geo-simulation, Sun Yat-sen University, Guangzhou 510006, China.

[2]School of Geospatial Engineering and Science, Sun Yat-sen University, and Southern Marine Science and Engineering Guangdong Laboratory (Zhuhai), Zhuhai 519082, China.

[3]Institute of Environment, Energy and Sustainability, The Chinese University of Hong Kong, Hong Kong SAR, China.

[4]School of Management, Guangdong University of Technology, Guangzhou 510520, China.

[5]Dalla Lana School of Public Health, University of Toronto, Toronto, Ontario M5T 3M7, Canada.

[6]School of Atmospheric Physics, Nanjing University of Information Science & Technology, Nanjing 210044, China.

[7]Tianjin Municipal Meteorological Observatory, Tianjin 300074, China.

[8]State Key Laboratory of Severe Weather, Chinese Academy of Meteorological Sciences, Beijing 100081, China.

[9]State Key Laboratory of Severe Weather (LASW) and Key Laboratory of Atmospheric Chemistry (LAC), Chinese Academy of Meteorological Sciences, Beijing 100081, China.

[10]College of Resources and Environment, Fujian Agriculture and Forest University, Fuzhou 35002, China.

*Correspondence to: Ming Luo ([luom38@mail.sysu.edu.cn](mailto:luom38@mail.sysu.edu.cn)) and Yongquan Zhao ([zhaoyq66@mail.sysu.edu.cn](mailto:zhaoyq66@mail.sysu.edu.cn))

## Abstract

Human-perceived thermal comfort (also known as human-perceived temperature) measures the combined effects of multiple meteorological factors (e.g., temperature, humidity, and wind speed) and can be aggravated under the influences of global warming and local human activities. With the most rapid urbanization and the largest population, China is being severely threatened by aggravating human thermal stress. However, the variations of thermal stress in China at a fine scale have not been fully understood. This gap is mainly due to the lack of a high-resolution gridded dataset of human thermal indices. Here, we generated the first high spatial resolution (1 km) dataset of monthly human thermal index collection (HiTIC-Monthly) over China during 2003–2020. In this collection, 12 commonly-used thermal indices were generated by the Light Gradient Boosting Machine (LGBM) learning algorithm from multi-source data, including land surface temperature, topography, land cover, population density, and impervious surface fraction. Their accuracies were comprehensively assessed based on the observations at 2419 weather stations across the mainland of China. The results show that our dataset has desirable accuracies, with the mean $R^2$, root mean square error, and mean absolute error of 0.996, 0.693°C, and 0.512°C, respectively, by averaging the 12 indices. Moreover, the data exhibit high agreements with the observations across spatial and temporal dimensions, demonstrating the broad applicability of our dataset. A comparison with two existing datasets also suggests that our high-resolution dataset can describe a more explicit spatial distribution of the thermal information, showing great potentials in fine-scale (e.g., intra-urban) studies. Further investigation reveals that nearly all thermal indices exhibit increasing trends in most parts of China during 2003–2020. The increase is especially significant in North China, Southwest China, the Tibetan Plateau, and parts of Northwest China, during spring and summer. The HiTIC-Monthly dataset is publicly available from Zenodo at https://zenodo.org/record/6895533 and the National Tibetan Plateau Data Center (TPDC) of China at https://data.tpdc.ac.cn/disallow/036e67b7-7a3a-4229-956f-40b8cd11871d (Zhang et al., 2022a).

## 1 Introduction

Global climate change has brought significant challenges to human society and natural systems (Arias et al., 2021; Haines and Ebi, 2019) by inducing higher air temperature and more frequent extreme weather and climate events around the world (Arias et al., 2021; Schwingshackl et al., 2021). Heat-related disasters, e.g., heatwaves, droughts, and wildfires, are occurring more frequently and becoming more intense (Tong et al., 2021; Arias et al., 2021; Luo et al., 2022), exacerbating the thermal environment and threatening the tolerance limits of humans, animals, and plants (Raymond et al., 2020). Substantial warming and increasing extreme weather and climate events aggravate human thermal comfort and increase the exposures to uncomfortable thermal environments (Brimicombe et al., 2021), thus posing adverse impacts on public health, socio-economy, and agricultural productivities (Budhathoki and Zander, 2019; Moda et al., 2019; Tuholske et al., 2021; Sun et al., 2019; Zhao et al., 2017).

The thermal stress that human beings actually perceive is not only related to air temperature, but also jointly influenced by other environmental variables such as humidity, wind, and/or direct sunlight (Mistry, 2020; Djongyang et al., 2010). These variables alter the heat balance that maintains the core temperature of human bodies by influencing the heat exchange (e.g., radiation, convection, conduction, and evaporation) between humans and the surrounding environment (Periard et al., 2021; Stolwijk, 1975). High atmospheric humidity can exacerbate the thermal stress on human bodies by reducing evaporation from the skin through sweating when the air temperature is high (Li et al., 2018; Rogers et al., 2021; Luo and Lau, 2021). Furthermore, abnormal weather with a combination of extremely high air temperature, humidity, and/or wind can reduce labor capacity and human performance (Roghanchi and Kocsis, 2018; Lazaro and Momayez, 2020; Enander and Hygge, 1990), leading to temperature-related discomfort, stress, morbidity, and even death (Di Napoli et al., 2018; Kuchcik, 2021; Nastos and Matzarakis, 2011), particularly during heatwaves. For example, in the summer of 2017, 2018, and 2019, there were 1489, 1700, and 161 heatwave-related deaths, respectively, in the United Kingdom (Rustemeyer and Howells, 2021). Additionally, vulnerable groups including children, the elderly, chronic patients, and poor communities are at higher risk of being affected by thermal stress (Patz et al., 2005; Wang et al., 2019), which is likely to be further exacerbated as global population aging and climate warming (United Nations, 2017).

84

The changes and impacts of human thermal stress have attracted increasing attention in recent years

(Schwingshackl et al., 2021; Krzysztof et al., 2021; Li et al., 2018; Rahman et al., 2022; Ren et al., 2022;

Luo and Lau, 2021). For instance, Szer et al. (2022) estimated the impact of heat stress on construction

workers based on the Universal Thermal Climate Index (UTCI). Ren et al. (2022) and Luo and Lau (2021)

quantified the contribution of urbanization and climate change to urban human thermal comfort in China.

Schwingshackl et al. (2021) assessed the future severity and trend of global heat stress based on Coupled

Model Intercomparison Project phase 6 (CMIP6). These studies were mainly based on meteorological

stations or coarse-gridded data. However, the meteorological stations are sparsely distributed (Peng et

al., 2019), particularly in undeveloped and mountainous areas, which cannot reveal continuously spatial

distributions of air temperature and thermal stress conditions (He et al., 2021). Additionally, existing low

spatial resolution image products (Mistry, 2020; Di Napoli et al., 2020) cannot be applied to fine-scale

studies because they cannot provide information with spatial details and variations. However, the changes

in human thermal stress at a fine scale (e.g., 1 km×1 km) remain much less understood. This research

gap is mainly inhabited by the unavailability of a high spatial resolution (high-resolution) gridded dataset

of human thermal stress.

100

Although extensive studies have been conducted to generate high-resolution land surface temperature

(LST) [such as the Land Surface Temperature in China (LSTC; (Zhao et al., 2020) and the global

seamless land surface temperature dataset (Zhang et al., 2022b; Hong et al., 2022)], or near surface air

temperatures (SAT) products [such as ERA5 (ECMWF, 2017), TerraClimate (Abatzoglou et al., 2018),

and GPRChinaTemp1km (He et al., 2021)], human thermal stress datasets were generally produced at

low-resolution levels, such as ERA5-HEAT (Di Napoli et al., 2020), HDI_0p25_1970_2018 (hereafter,

HDI) (Mistry, 2020), and HiTiSEA (Yan et al., 2021). ERA5-HEAT was derived from ERA5 and

includes two global hourly human thermal stress indices (UTCI and mean radiant temperature (MRT))

from January 1979 to the present (Di Napoli et al., 2020). The HDI dataset was generated using 3-hourly

climate variables of the global land data assimilation system (GLDAS), and it contains ten daily indices

with a spatial resolution of $0.25° \times 0.25°$, covering 90°N–60°S from 1970 to 2018 (Mistry, 2020).

HiTiSEA contains ten daily human thermal stress indices from 1981 to 2017, with a spatial resolution of

0.1° × 0.1° over South and East Asia (Yan et al., 2021), which was derived from the ERA5-Land and
ERA5 reanalysis products. However, these existing thermal index datasets have very coarse spatial
resolutions. There is an urgent need for a high-resolution (e.g., 1 km) data collection of multiple human
thermal stress indices.

Various indices have been proposed to measure human thermal stress, but there is no universal thermal
stress index that works in all climate zones (Schwingshackl et al., 2021; Brake and Bates, 2002;
Roghanchi and Kocsis, 2018; Luo and Lau, 2021). Existing human thermal stress indices considered
different climate conditions, direct or indirect exposures to weather elements, human metabolism, and
the local working environment (Di Napoli et al., 2020), which were designed to evaluate or quantify the
comprehensive environmental pressure of meteorological factors (e.g., temperature, humidity, wind) on
human bodies (Epstein and Moran, 2006). These indices are based on the thermal exchange between the
human and surrounding environments or empirical relationships gained by studying human responses to
various environmental factors, varying in complexity, applicability, and capacity (Staiger et al., 2019).
For example, the heat index (HI) is used for meteorological service (NWS, 2011); wet-bulb temperature
(WBT) is used to measure the upper physiological limit of human beings (Raymond et al., 2020);
physiologically equivalent temperature (PET) and UTCI are used to estimate human thermal comfort
(Varentsov et al., 2020). Therefore, a high-resolution dataset that contains different commonly used
human thermal stress indices is urgently called for in global and regional studies, particularly for those
with complex climate conditions (e.g., China).

China has been threatened by deteriorating thermal environments under global climate change and rapid
local urbanization over the past decades (Ren et al., 2022; Luo and Lau, 2019). The changes and
characteristics of human thermal stress across China have attracted extensive attention in recent years
(Yan, 2013; Tian et al., 2022; Li et al., 2022). Wang et al. (2021) found that the frequency of extreme
human-perceived temperature events increases in summer and decreases in winter in most urban
agglomerations (UAs) of China. Li et al. (2022) showed that the frequency of thermal discomfort days
in China exhibits a significant increasing trend from 1961 to 2014, and there will be more threats from
thermal discomfort in the future. Therefore, a long-term and high-resolution dataset with multiple human

thermal stress indices in China is of great importance for investigating detailed spatial and temporal variations of human thermal stress across the country. Such a dataset has the potential to (1) assess population exposure to extreme thermal conditions and heat-related health risks, (2) reveal the spatiotemporal evolution of human thermal stress and its influence on public health, tourism, industries, military, epidemiology, and biometeorology at a fine scale, and (3) provide policymakers with data in manipulating targeted strategies to mitigate heat stress and protect vulnerable people.

In this study, we produced a high-resolution (1 km × 1 km) thermal index collection at a monthly scale (HiTIC-Monthly) in China over a long period (2003–2020). This collection contains 12 widely-used human thermal indices, including Surface Air Temperature (SAT), indoor Apparent Temperature ($AT_{in}$), outdoor shaded Apparent Temperature ($AT_{out}$), Discomfort Index (DI), Effective Temperature (ET), Heat Index (HI), Humidex (HMI), Modified Discomfort Index (MDI), Net Effective Temperature (NET), Wet-Bulb Temperature (WBT), simplified Wet Bulb Globe Temperature (sWBGT), and Wind Chill Temperature (WCT). The remainder of this paper is structured as follows. Sections 2 and 3 describe the data sources and the methodology, respectively. Section 4 presents a comprehensive analysis of the accuracies and trends of the human thermal indices. Comparisons on our products with two existing datasets are in Section 5, data availability is provided in Section 6. The main findings of this paper are summarized in Section 7.

## 2 Data

### 2.1 Meteorological data

Daily mean surface air temperature, relative humidity, and wind speed recorded at the 2419 weather stations across China (Figure 1) during 2003–2020 were collected from the China Meteorological Data Service Center (CMDC) at http://data.cma.cn/en. All station records were subjected to strict quality control and evaluation, including homogenization based on a statistical approach (Xu et al., 2013) and evaluation of temporal inhomogeneity based on the Easterling-Peterson method (Li et al., 2004).

**2.2 Covariates**

Human thermal stress is related to temperature, topography, land cover, population density, surface water, and vegetation (Wang et al., 2020; Rahman et al., 2022; Krzysztof et al., 2021). In this study, eight variables reflecting the changes and spatial distribution characteristics of temperature were used to predict human thermal indices (Table 1) in addition to the meteorological variables. As LST is one of the most essential parameters for predicting human thermal indices, the seamless LST dataset created by Zhang et al. (2022b) was introduced into our model training. This LST dataset used a spatiotemporal gap-filling algorithm to fill the missing or invalid value caused by clouds in the Moderate Resolution Imaging Spectroradiometer (MODIS) LST dataset (MOD11A1 and MYD11A1). It includes daily mid-daytime (13:30) and mid-nighttime (01:30) LST with 1 km spatial resolution. The mean root mean squared errors (*RMSEs*) of daytime and nighttime LST are 1.88°C and 1.33°C, respectively. We used monthly LST as one of the inputs to predict the spatial distribution of 12 thermal indices. Monthly LST values were calculated by averaging daily LST, which was obtained by averaging four observations in a day, including mid-daytime and mid-nighttime observations from ascending and descending orbits of MOD11A1 (Terra) and MYD11A1 (Aqua). More details about the LST data are described in Zhang et al. (2022b). The land cover dataset (MCD12Q1 Version 6) developed by Sulla-Menashe and Friedl (2019) based on a supervised classification method was downloaded via Google Earth Engine (GEE). The Multi-Error-Removed Improved-Terrain (MERIT) elevation dataset developed by Yamazaki et al. (2017) was downloaded from GEE. This dataset was generated after removing the errors from existing Digital Elevation Models (DEMs), such as SRTM3 and AW3D-30m, based on multi-source satellite data and filtering algorithms. The spatial resolution of this dataset is 3″ (i.e., ~90 meters at the equator). In addition, the slope was also extracted from the elevation data to act as the topography predictor. As the artificial surface is closely related to human activities (Zhao and Zhu, 2022), the dataset of global artificial impervious area (GAIA) produced by Gong et al. (2020) from the Google Earth Engine (GEE) was used to delineate human footprints. The overall accuracy of GAIA is greater than 90% (Gong et al., 2020). The population dataset was downloaded from the WorldPop Project (Gaughan et al., 2013). Then, the abovementioned eight datasets were pre-processed to have the same spatial extend, projection, and spatial resolution (1 km) through image mosaicking, reprojection, resampling, clipping, aggregating, and monthly synthesizing. Moreover, year and month of the year were also used as covariates. Note that we

did not include precipitation as a covariate because the precipitation data are not normally distributed. More importantly, they exhibit many zero values in many regions of China (especially in the dry season), which would increase the uncertainty of the spatial prediction.

## 3 Methodology

### 3.1 Calculation of human thermal indices

In addition to SAT, the calculation of human thermal indices used in this study is described in Table 2. These indices are first calculated based on SAT (also simply denoted as T), relative humidity (RH), wind speed (V), and actual vapor pressure ($E_a$) at daily scale. $E_a$ is derived from T and RH rather than directly observed at meteorological stations (Eqs. 1~2; (Bolton, 1980)). Furthermore, monthly human thermal indices were derived by averaging daily values in each month.

$$E_s = 6.112 \times exp^{(17.67 \times T/(T+243.5))} \tag{1}$$

$$E_a = \frac{RH}{100} \times E_s \tag{2}$$

Here $E_s$ is saturation vapor pressure (hPa) near the surface, T (°C) is air temperature at 2 m above the ground, and RH (%) is relative humidity at 2 m above the ground.

### 3.2 Prediction of human thermal indices using LGBM

The Light Gradient Boosting Machine (LGBM) algorithm was employed to predict human thermal indices during 2003–2020. LGBM is one of the gradient boosting decision tree (GBDT) algorithms developed by Microsoft Research (Ke et al., 2017). This algorithm has become a very popular nonlinear machine learning algorithm due to its superior performance in machine learning competitions and efficiency (Candido et al., 2021). Its performance has been evaluated and shows desirable results in different applications, such as evapotranspiration estimation (Fan et al., 2019), land cover classification (Candido et al., 2021; Mccarty et al., 2020), air quality prediction (Su, 2020; Zeng et al., 2021; Tian et al., 2021), subsurface temperature reconstruction (Su et al., 2021), and above-ground biomass estimation (Tamiminia et al., 2021).

Furthermore, LGBM adopts the Gradient-based One-Side Sampling (GOSS) and Exclusive Feature Bundling (EFB) algorithms to improve the training speed (Su et al., 2021). Here, GOSS is used to select data instances with larger gradients and to exclude a considerable proportion of small gradient data instances (Ke et al., 2017), and EFB is used to merge features (Ke et al., 2017). Compared with traditional GBDT algorithms including eXtreme gradient boosting (XGBoost) and Stochastic Gradient Boosting (SGB), LGBM effectively decreases the training time without reducing the accuracy (Los et al., 2021; Ke et al., 2017).

We used the Python package *Scikit-Learn* to perform the LGBM training, and hyperparameters of LGBM were tuned based on Grid Search Methods. The observed monthly human thermal indices at the 2419 weather stations across the mainland of China during 2003–2020 were randomly classified into a training set (80%) for hyperparameters tuning and model training and a testing set (20%) for model evaluation.

**3.3 Accuracy assessment**

Four statistic metrics, namely, determination coefficient ($R^2$), Mean Absolute Error (*MAE*), *RMSE*, and *Bias* (Rice, 2006), were used to evaluate the prediction accuracy of the human thermal indices. Ranging from 0 to 1, $R^2$ measures the proportion of variance explained by the model, representing how well the human thermal indices were predicted compared to the observations. *MAE* represents the average absolute error between the predictions and the observations. *RMSE* is the standard deviation of the residuals and is sensitive to outliers. *Bias* describes the differences between the predictions and the observations. These metrics are computed as follows.

$$MAE = \frac{1}{N} \times \sum_{i=1}^{N} |y_i - \hat{y}| \tag{3}$$

$$RMSE = \sqrt{\frac{1}{N} \times \sum_{i=1}^{N} (y_i - \hat{y})^2} \tag{4}$$

$$R^2 = 1 - \frac{\sum_{i=1}^{N}(y_i - \hat{y})^2}{\sum_{i=1}^{N}(y_i - \bar{y})^2} \tag{5}$$

$$Bias = \frac{1}{N} \times \sum_{i=1}^{N}(y_i - \hat{y}) \tag{6}$$

where $\hat{y}$ is the predicted value of human thermal indices, $\bar{y}$ is the mean of the observed human thermal indices calculated from meteorological stations, and $N$ is the number of samples.

251

## 4 Results

### 4.1 Evaluation of the predicted human thermal indices

#### 4.1.1 Overall accuracy

The prediction accuracies of the 12 human thermal indices were evaluated based on the validation data introduced in Section 3.2. All predicted human thermal indices exhibit high accuracies. Figure 2 shows the scatter plots of the observed versus the predicted values of the 12 human thermal indices. As the figure displays, the data points of all indices are concentrated around the corresponding 1:1 line, indicating a good consistency between the observed and the predicted values. Figure 3 and Table 3 present the $R^2$, *MAE*, *RMSE,* and *Bias* values of 12 thermal indices during 2003–2020. The $R^2$ values of the 12 indices are all higher than 0.99, and their *RMSE*, *MAE,* and *Bias* are lower than 0.9 °C, 0.7 °C, and 0.003 °C, respectively. Particularly, HMI has the largest *RMSE* (0.859 °C) and *MAE* (0.645 °C), while ET shows the smallest *RMSE* (0.377 °C) and *MAE* (0.281 °C). The larger errors of NET are likely caused by the incorporation of wind speed during the computation (see Table 2). Overall, the accuracy metrics demonstrate that the 12 predicted human thermal indices are of good quality.

The spatial distributions of $R^2$, *MAE, RMSE*, and *Bias* at individual stations across the mainland of China are depicted in Figure 4–7, respectively. The predicted indices have high $R^2$ values (i.e., >0.98, Figure 4) at almost all stations across China, demonstrating the superiority of LGBM. Better predictions (with higher $R^2$) are distributed in eastern China, particularly in the North China Plain (NCP) and the Yangtze River Delta (YRD), while southwestern China (e.g., the Yunnan-Guizhou Plateau (YGP)) has relatively lower $R^2$ values (<0.98). For *MAE* and *RMSE*, all indices have small values <1 °C at most stations across China. HMI has the largest *MAE* and *RMSE* values (Figure 5g and 6g), followed by NET and WCT, and ET has the smallest *MAE* and *RMSE* values (i.e., < 0.4 °C, Figure 5e and 6e). The *MAE* and *RMSE* of NET and WCT decrease from northwestern to southeastern China (Figure 5i, 5l, 6i, 6l). For other indices, small *MAE* and *RMSE* values are mainly observed in plains including NCP, while large values tend to appear in regions with complex topography, such as arid Northwest China, mountainous Northeast and South China, and the

Hengduan Mountains. These differences are related to the uneven distribution of weather stations,
i.e., dense in plains and coarse in complex terrain areas. The *Bias* values range from -0.3 °C to 0.3 °
C (Figure 7). Positive *Bias* values tend to distribute in northern China while negative values are
mainly located in the south. This spatial variability is likely caused by the generally lower
temperatures in the north and higher temperatures in the south. In particular, the extremely small
values in the north and the extremely large values in the south may be overestimated and
underestimated to some extent, respectively, due to limited samples of extremely small and large
values (compared with the rest of the samples) when training the machine learning model. The
overestimation and underestimation issues caused by limited training samples of extreme values are
quite common in machine learning (Wu et al., 2022; Li et al., 2020; Uddin et al., 2022; Cho et al.,

289     2020).


**4.1.2 Annual and monthly accuracies**
The annual accuracies regarding *RMSE, MAE,* and *Bias* of the 12 human thermal indices during 2003–
2020 are shown in Figure 8. *RMSEs* and *MAEs* of all indices in nearly all years are less than 1.0 °C
(Figure 8a-b). Yearly *RMSE* (*MAE*) of ET fluctuates around 0.3 °C (0.2 °C) during 2003–2020. *RMSEs*
(*MAEs)* of other indices range from 0.5 to 1.1 °C (0.4–0.8 °C) with marginal variations from year to year.
*Biases* vary between -0.04 °C and 0.04 °C across all years. This temporal variability of the *Bias* is related
to the yearly climate variations, and is characterized by a marginal overestimation of lower temperatures
that mainly appeared in early periods (e.g., 2003–2005) and the underestimation of higher temperatures
mostly in recent periods (e.g., 2016–2019). Under climatic warming over the past decades, the lower
temperatures tended to appear in early periods while relatively higher temperatures more likely occurred
in more recent periods. Extremely small values of temperature in earlier periods and the large values in
the later periods may be slightly overestimated (i.e., with positive *Bias* values) and underestimated (i.e.,
with negative *Bias* values), respectively, thereby characterizing the temporal variations of the *Bias*.
Moreover, Figure S1 displays the monthly *RMSEs*, *MAEs,* and *Biases* of all human thermal indices. For
*RMSE*, all the indices in 12 months are lower than 1.4 °C, and their *MAEs* are less than 1 °C. HI and HMI
have relatively higher *RMSE* and *MAE* values in summer than in other seasons; whereas, other indices
tend to have larger errors in winter than in summer. Additionally, the magnitude of *Bias* is smaller than
0.03 °C for all the indices in 12 months.

**4.1.3 Accuracies in major urban agglomerations**
More than half of the national population in China lives in cities, particularly in UAs (i.e., also known as
city clusters). Here we assessed the prediction accuracies in 20 major UAs in China, which hold 62.83%
and 80.57% of the total population and gross domestic product (GDP) of the country (Fang, 2016). These
accuracy assessments are presented in Tables S1–S4. As shown in Table S1, all UAs have $R^2$ values
higher than 0.9837, with an average of 0.9947. Table S2 also shows that these UAs have small *RMSE*
values, most of which are smaller than 1 °C, except for the UA of North Tianshan Mountain in arid
Northwest China. As the biggest UA in China, YRD has the lowest *RMSE* of 0.288 °C among all 20 UAs.
The *MAEs* of the thermal indices in all UAs are smaller than 1 °C and with an average value of 0.477 °C
(Table S3). The *Biases* in the 20 UAs range from -0.160 °C to 0.123 °C (Table S4). These results suggest
that all predicted human thermal indices in different UAs across China are of good quality at the local
scale. It implies that our prediction model and results have great potential in evaluating local thermal
environment changes (e.g., in urban areas or cities).

**4.2    Spatial variations of the human thermal indices**
The abovementioned assessments show that our model based on LGBM can yield high-accuracy
predictions at both national and local scales. Therefore, this model is employed to generate a high-
resolution human thermal index collection at a monthly scale over China (HiTIC-Monthly) during 2003–
2020. By taking monthly ET in 2020 as an example, we examined the monthly evolution of spatial
patterns of the HiTIC-Monthly dataset in this subsection.

Figure 9 shows the monthly distribution of the predicted ET in 2020, which exhibits obvious seasonality
with higher temperatures in summer and lower in winter. The temperature shows a significant zonal
difference with colder temperatures in northern than in southern China. The temperature has a close
relationship with topography and decreases with elevation, varying from plateaus to plains. The Qinghai-
Tibet Plateau (TP) has the lowest temperature, while southern China, the Sichuan Basin, and the Gobi

regions in Northwest China witness the highest temperature. The distribution of temperature exhibits different patterns among the four seasons, especially between winter (e.g., January) and summer (e.g., July). In winter, the temperature increases from northern to southern areas and is the coldest in Northeast and Northwest China and the warmest on the Hainan Island. In summer, the hottest temperature appears in the Tarim and Jungar Basins of Xinjiang. The NCP region also has a high temperature in summer, which might be related to local urbanization (Liu et al., 2008) and irrigation (Kang and Eltahir, 2018).

The spatial variations of the predicted human thermal indices in summer (which is often characterized by severe heat stress) are examined in Figure 10 by taking July 2020 as an example. As it shows, the 12 indices exhibit similar distribution patterns. There are significant differences in temperature among northwest, northeast, and southeast China. Generally, the temperature decreases from the southeast to the northwest, and the southeast and northwest parts have the highest and lowest temperatures, respectively.

HMI exhibits the highest temperature while NET shows the lowest in July 2020. The dominant modes of these indices are further examined by applying the empirical orthogonal function (EOF) analysis (Figures S10–S13). As Figure S10 shows, the leading EOF (EOF1) of all 12 thermal indices exhibit highly consistent spatial distribution with higher values in the northern region and lower values in the south. Their temporal variations are also similar to each other (Figure S11). The second and third EOF modes (EOF2 and EOF3) are also similar among different thermal indices (except EOF3 of NET, Figures S11–S13). These results demonstrate the desirable quality of our products.

**4.3 Temporal changes in the human thermal indices**

The yearly evolutions of the annual mean human thermal indices during 2003–2020 are displayed in Figure 11. Despite the interannual fluctuation in the time series, all indices exhibit upward trends except for NET and WCT, of which the decreasing trends are mainly affected by the recovering wind speed in the recent decade (Zeng et al., 2019). The fastest warming appears in HMI (0.303 °C/decade), and the slowest is in ET (0.111 °C/decade). These warming trends are stronger than the rising rate of global mean near surface temperature (IPCC, 2021), demonstrating China as one of the severest hotspots suffering from dramatic climate warming under global change. The detailed spatial variations regarding the trends

of the human thermal indices across China are further depicted in Figure 12. Most parts of China
experience are seen with increases in nearly all the indices during 2003–2020. These increases are
especially more profound in North China, Southwest China, TP, and parts of Northwest China. The
possible reasons for the prominent warming trends in North China are explained as follows. The
urbanization process has been prevailing in this area, with rapid growth in the economy and population.
This process is accompanied by dramatic increases in impervious surfaces and decreases in green spaces.
These changes lead to warmer surface and near surface air temperature, known as urban heat islands
(UHI), thus increasing thermal stress in this region. The urbanization effects on local heat stress have
also been reported by (Luo and Lau, 2021). Moreover, North China has a large amount of croplands with
prominent irrigation activities, which may increase air humidity near the surface and exacerbate the
combined effects of temperature and humidity, leading to increased heat stress (Kang and Eltahir, 2018).
In addition, this area has experienced a weakening of surface wind speed (Zhang et al., 2021), which also
exacerbates thermal stress, especially in NET and WCT.

Furthermore, different indices have different degrees of increasing trends. HMI has the largest
increasing magnitude (Figure 12h), and ET is seen with relatively slight increases across China
(Figure 12f). The trends of NET and WCT have similar spatial distribution patterns, with large
proportions having cooling trends since 2003 (Figure 12j&l). Most parts of Xinjiang, northeastern
and southern China have obvious decreasing trends, and the Inner Mongolia Plateau (IMP), NCP,
eastern TP, YRD, and YGP have slightly increasing trends.

The temporal trends of the human thermal indices in different seasons were also examined (Figure 13).
The fastest warming tendency is observed in the spring season. The rising trends of spring HMI, HI, MDI,
$AT_{in}$, and $AT_{out}$ exceed 0.4 °C/decade, and the trends of other indices (except ET and NET) are larger
than 0.3 °C/decade (Figure S2). Summer also has been experiencing significant increasing trends in all
indices, i.e., at a rate of > 0.2 °C/decade (except ET and NET). The trends in summer HMI, HI, WBT,
MDI, DI, sWBGT, $AT_{in}$, and $AT_{out}$ exceed 0.3 °C/decade (Figure S3). Differing from spring and summer,
the human thermal indices (except WCT and NET) in the autumn season show slightly cooling trends
(Figure S4). Autumn WCT and NET have significantly strong decreasing trends, i.e., -0.349 and -

0.507 °C/decade, respectively. Similar strong cooling trends of WCT and NET appear in winter, i.e., -0.661 and -0.453 °C/decade, respectively, while other indices experience marginal long-term changes (Figure S5).

Figure S6 maps the spatial patterns of the trends of summer mean human thermal indices over the mainland of China during 2003–2020. All indices show warming trends in most parts of China, particularly in NCP and TP. As one of the most densely populated regions in China, the prominent increases in thermal indices in NCP indicate that the local has been experiencing increasing threats of intensifying heat stress. Among the 12 indices, $AT_{out}$, HI, NET and WCT tend to have a slight cooling trend in southeastern China. This cooling trend is consistent with the corresponding summer SAT.

The spatial distributions of the changing trends in winter across the mainland of China during 2003–2020 are depicted in Figure S7. The trend patterns in winter are similar to that in summer to some degree. The warming trends are concentrated in Southwest China, most parts of Northwest China, and parts of East China (e.g., YRD). The cooling trends are located in TP, parts of Northeast and South China. The cooling tendencies are especially significant in Northeast China, and most parts of Northwest and South China (Figures S7 j&m). Parts of central China are seen with even stronger cooling thermal comfort.

In spring, increases in all thermal indices are observed in most parts of China (Figure S8), particularly in northern regions, such as central Inner Mongolia, parts of NCP, and Northeast China, while parts of southern China have slight decreases. These decreases are noticeable in NET and WCT (Figures S8 j&m). In contrast to spring, the autumn season is observed with decreased thermal temperature in the north and increases in the south (e.g., Southwest China, Figure S9).

## 5 Discussion

### 5.1 Comparison with existing human thermal index datasets

We compared our HITIC-Monthly with two existing datasets, i.e., HDI (Mistry, 2020) and HiTiSEA (Yan et al., 2021), which have coarser spatial resolutions of 0.25°×0.25° and 0.1°×0.1° (Table 4), respectively.

We derived monthly mean $AT_{in}$ in July 2018 from HDI and HiTiSEA and compared them with HITIC-
Monthly over the mainland of China, with a particular highlight in the four largest UAs, including
Beijing-Tianjin-Hebei (BTH), YRD, middle Yangtze River Valley (mYRV) and Pearl River Delta (PRD)
(Figure 14). The summer of 2018 was selected because it was included in all three datasets and frequent
heat events occurred in this summer (Zhou et al., 2020). Generally, the three datasets depict similar spatial
patterns. However, our HiTIC-Monthly dataset obviously provides more detailed and clearer spatial
information on human thermal stress than the other two. Additionally, the observed $AT_{in}$ values at
individual weather stations are also compared (Figure 14). It can be seen that HDI and HiTISEA
overestimate $AT_{in}$, and such overestimation is especially severe for HDI, while our dataset is in good
agreement with the observed $AT_{in}$ at individual weather stations. Therefore, our predicted temperature
can describe the spatial variations in the city areas well, thereby providing fundamental support for fine-
scale climate studies, such as urban climate research.
**5.2 Limitations and future works**
There are 12 commonly used human thermal indices in the HiTIC-Monthly dataset produced in this study.
Nine of these indices were computed from temperature and humidity (or water vapor) and the other three
(i.e., $AT_{out}$, NET, and WCT) were derived from temperature, humidity, and wind speed. In addition, other
indices considering the combined effect of environmental variables such as sunlight (Blazejczyk, 1994;
Fanger, 1970; Höppe, 1999; Yaglou and Minaed, 1957) were proposed, including wet bulb globe
temperature (WBGT), predicted mean vote (PMV), UTCI, physiological equivalent temperature (PET),
etc. These thermal indices were not included in our study due to the lack of sunshine and radiative flux
data.

Since LST is the most important variable for predicting the 11 human thermal indices, the uncertainty in
the LST dataset may influence the accuracy of the human thermal indices. The LST variable in our
prediction is collected from a global seamless 1 km resolution daily LST dataset (Zhang et al., 2022b).
This dataset was generated based on spatiotemporal gap-filling algorithms and the MODIS LST data. It
may overestimate LST in some cases because the LST under cloudy weather was filled based on the data
in clear sky conditions (Zhang et al., 2022b). A high-quality LST dataset would further improve the
prediction accuracy of the human thermal indices.

The human thermal indices dataset is at a monthly scale, but the temporal resolution may not be sufficient

for the research of extreme weather events (e.g., heatwaves and cold spells) and related environmental

health (e.g., heat-related mortality). A daily high-resolution human thermal index collection (HiTIC-

Daily) will be produced and released in our future studies. In the current study, we provided the first

national-level dataset over the mainland of China with multiple high-resolution human thermal indices

in a monthly interval, which shows high prediction accuracies in all climate regimes across China. A

global dataset of multiple human thermal indices dataset is also expected in the near future.

## 6 Data availability

The high spatial resolution monthly human thermal index collection (HiTIC-Monthly) generated in

this study is freely available to the public in network common data form (NetCDF) from Zenodo at

https://zenodo.org/record/6895533 and the National Tibetan Plateau Data Center (TPDC) of China at

https://data.tpdc.ac.cn/disallow/036e67b7-7a3a-4229-956f-40b8cd11871d (Zhang et al., 2022a). The

human thermal indices include surface air temperature (SAT), indoor Apparent Temperature ($AT_{in}$),

outdoor shaded Apparent Temperature ($AT_{out}$), Discomfort Index (DI), Effective Temperature (ET),

Heat Index (HI), Humidex (HMI), Modified Discomfort Index (MDI), Net Effective Temperature

(NET), simplified Wet Bulb Globe Temperature (sWBGT), Wet-Bulb Temperature (WBT), and

Wind Chill Temperature (WCT). This dataset has a spatial resolution of 1 km×1 km and covers the

mainland of China from 2003 to 2020, stacking by year. Each stack is composed of 12 monthly

images. The unit of the dataset is 0.01 degree Celsius (°C), and the values are stored in an integer

type (Int16) for saving storage space, and need to be divided by 100 to get the values in degree

Celsius when in use. The projection coordinate system is Albers Equal Area Conic Projection. The

naming rule and other detailed information can be found in "README.pdf".

## 7 Conclusions

A long-term and high-resolution dataset of multiple human thermal indices is of great significance for

monitoring detailed spatiotemporal changes of human thermal stress in different climate regions across
China and assessing the health risks of people exposed to extreme heat at a fine scale. However, the
current datasets of human thermal indices (e.g., HDI and HiTiSEA) only have coarse spatial resolutions
(> 0.1°). In this study, we generated a dataset of monthly human thermal index collection with a high
spatial resolution of 1 km over the mainland of China (HiTIC-Monthly). In this collection, 12 human
thermal indices from 2003 to 2020 were predicted, including SAT, $AT_{in}$, $AT_{out}$, DI, ET, HI, HMI, MDI,
NET, sWBGT, WBT, and WCT.

The HiTIC-Monthly dataset was produced by LGBM based on multi-source data, including MODIS LST,
DEM, land cover, population density, and impervious surface fraction. This dataset shows a desirable
performance, with mean $R^2$, *RMSE*, *MAE,* and *Bias* of 0.996, 0.693°C, 0.512°C, and 0.003°C,
respectively. Our predictions also exhibit good agreements with the observations in both spatial and
temporal dimensions, demonstrating the broad applicability of our dataset. Moreover, the comparison
with two existing datasets (i.e., HDI and HiTiSEA) suggests that HiTIC-Monthly has more detailed
spatial information, indicating that our dataset can well support fine-scale studies. Further investigation
shows that almost all the indices show warming trends in most parts of China during 2003–2020,
particularly for North China, Southwest China, TP, and parts of Northwest China. Additionally, the
warming tendency is faster in spring and summer. WCT and NET show similar and strong cooling trends
in autumn and winter, while other indices exhibit slight long-term changes.

## Author contribution

H.Z.: Data curation, Formal analysis, Investigation, Methodology, Writing – original draft preparation;
M.L.: Formal analysis, Conceptualization, Investigation, Funding acquisition, Methodology, Supervision
Writing – review & editing; Y.Z.: Formal analysis, Conceptualization, Investigation, Supervision,
Writing – review & editing; L.J.: Investigation, Writing – review & editing; E.G.: Investigation, Writing
– review & editing; Y.Y.: Investigation, Writing – review & editing; G.N.: Investigation, Writing – review
& editing; J.G.: Investigation, Writing – review & editing; Z.Z.: Investigation, Writing – review & editing;
K.G.: Investigation, Writing – review & editing; J.L.: Investigation, Writing – review & editing; X.L.:
Investigation, Writing – review & editing; S.W.: Investigation, Writing – review & editing; P.W.:
Investigation, Writing – review & editing; X.W.: Investigation, Writing – review & editing.

## Competing interests

The authors declare that they have no conflict of interest.

## Acknowledgments

This work was supported by the National Natural Science Foundation of China (41871029), the
Guangdong Basic and Applied Basic Research Foundation (2019A1515011025), the National Youth
Talent Support Program of China, the Pearl River Talent Recruitment Program of Guangdong Province
(2017GC010634), and the Innovation Group Project of Southern Marine Science and Engineering
Guangdong Laboratory (Zhuhai) (311021008). The authors are grateful to the editor and two reviewers
whose comments and suggestions have significantly improved the quality of our manuscript.

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

**Figures**

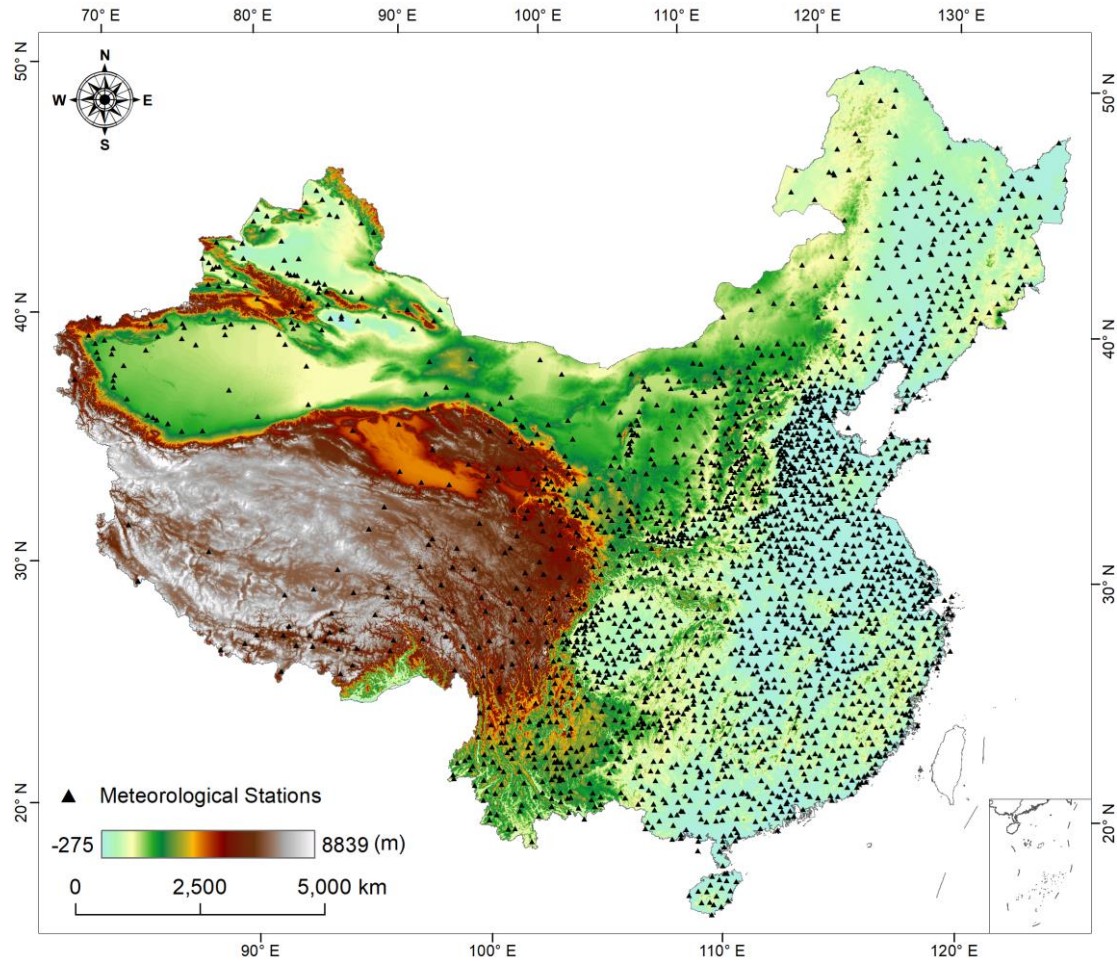


**Figure 1. Spatial distribution of meteorological stations in the mainland of China, with color shadings**
**indicating the elevation in meters.**

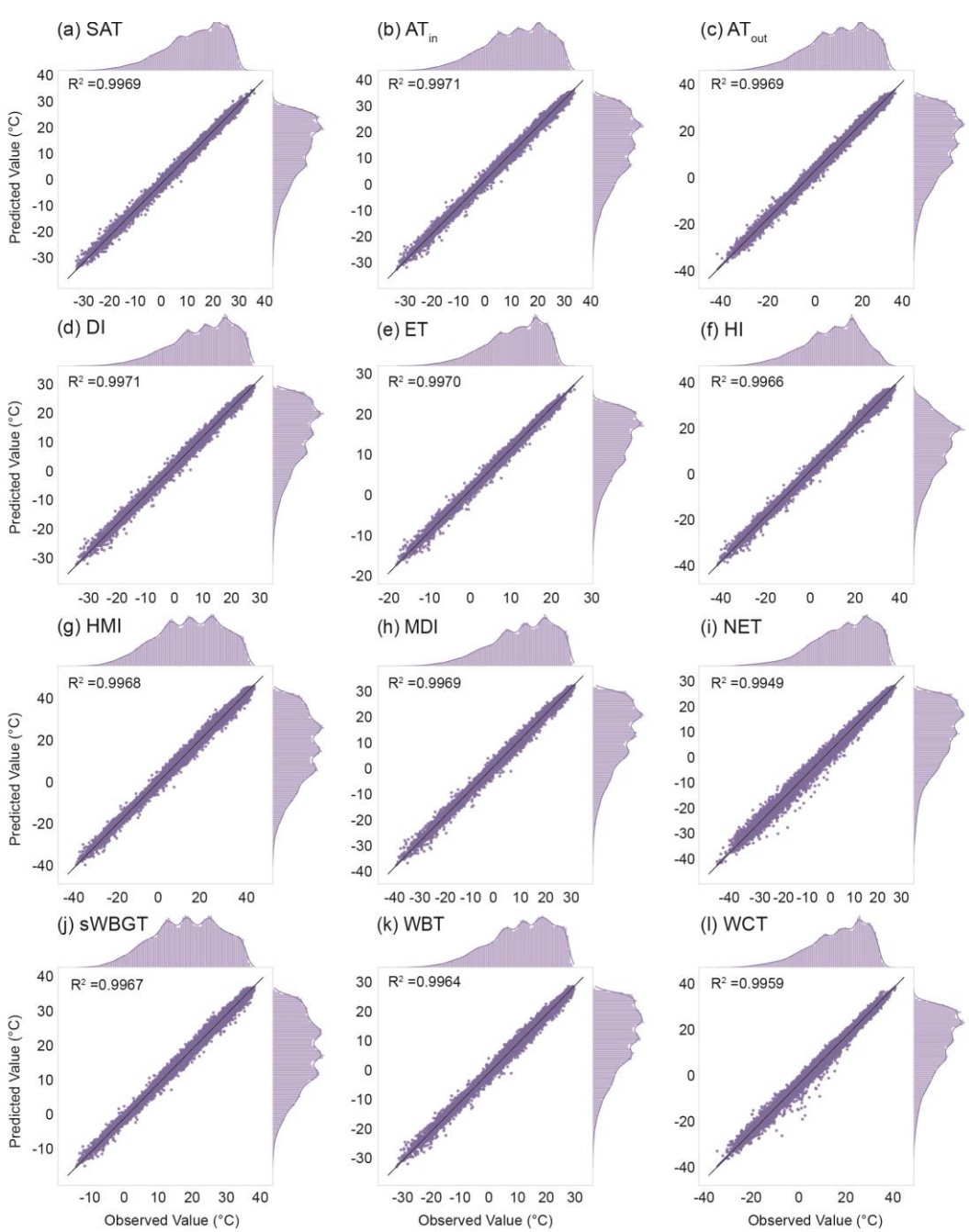

**Figure 2. Scatter plots of predictions versus observations of the 12 human thermal indices over the mainland**
**of China during 2003–2020. (a) SAT, (b) AT$_{in}$, (c) AT$_{out}$, (d) DI, (e) ET, (f) HI, (g) HMI, (h) MDI, (i) NET, (j)**
**sWBGT, (k) WBT, and (l) WCT.**

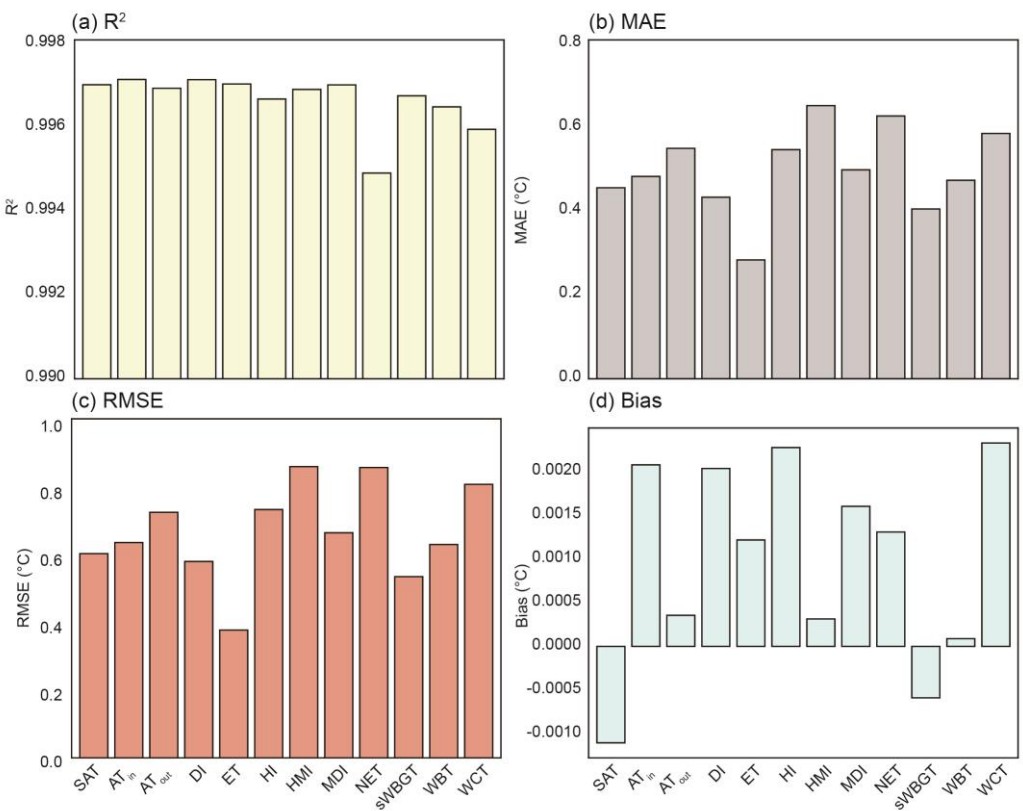


**Figure 3. Overall prediction accuracies of the 12 human thermal indices over the mainland of China during**
**2003–2020. (a)** $R^2$**, (b)** *MAE***, (c)** *RMSE***, (d)** *Bias***.**

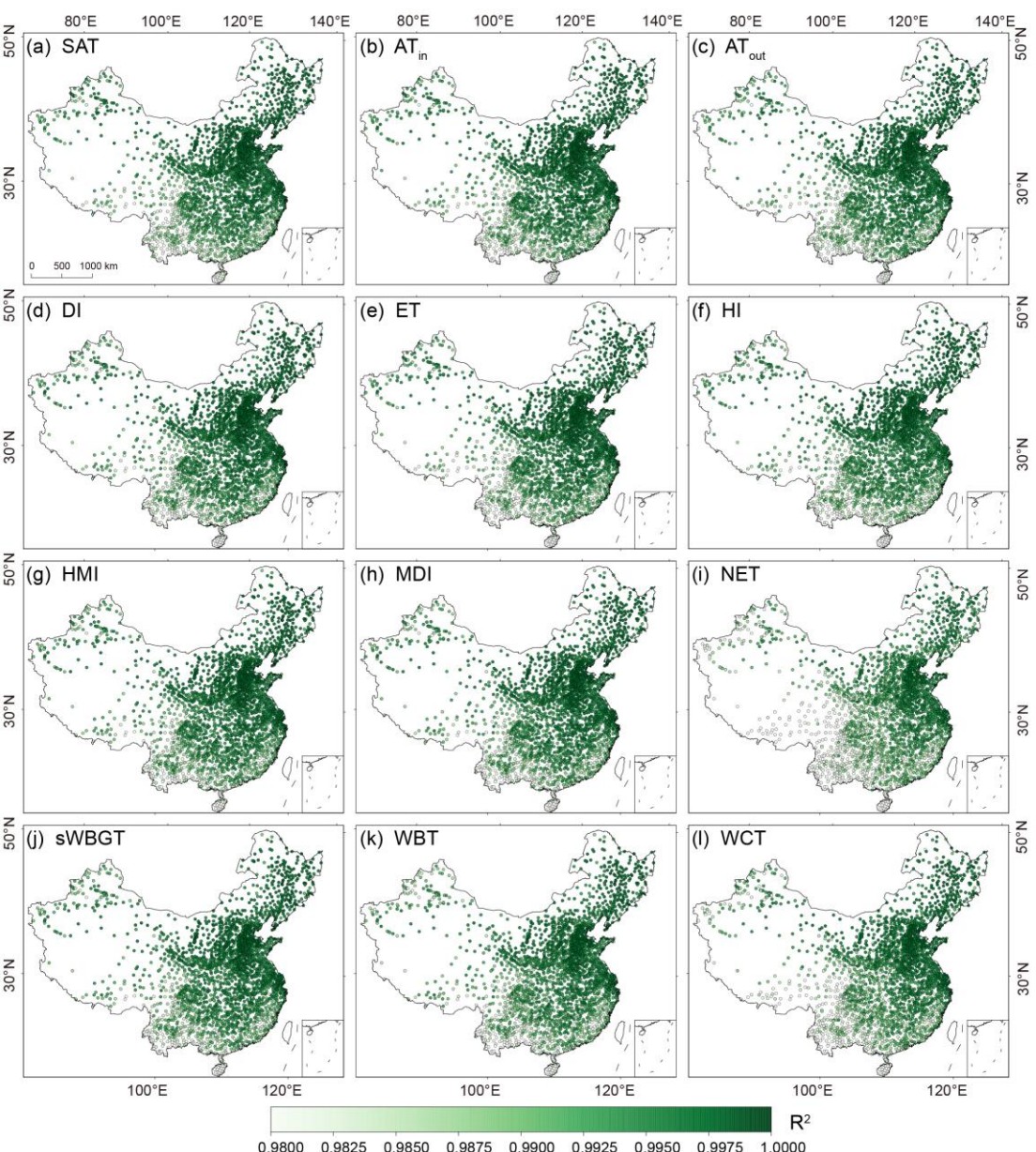


**Figure 4. Spatial distribution of $R^2$ of the 12 human thermal index predictions at individual meteorological stations over the mainland of China during 2003–2020.**


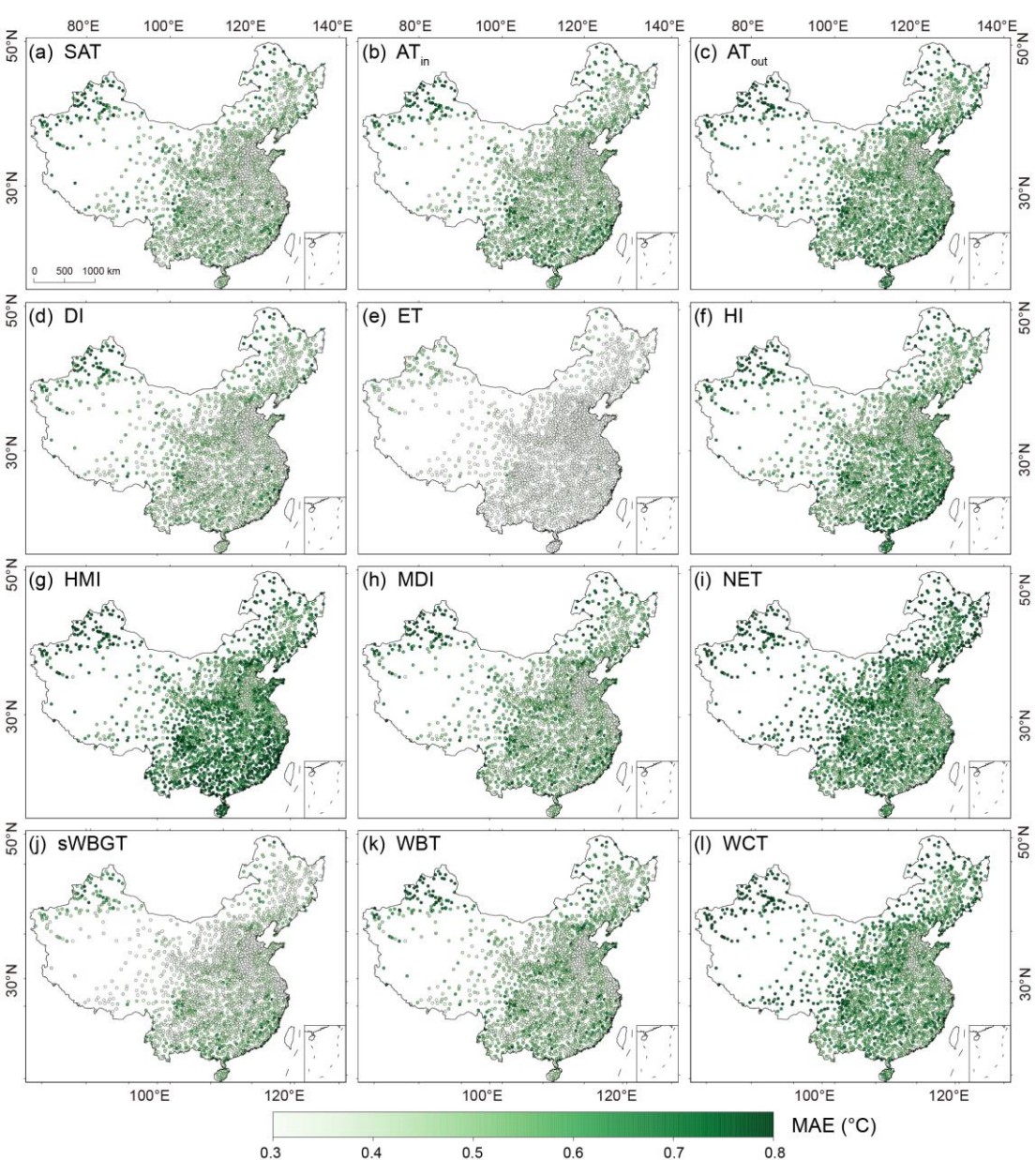


**Figure 5. As Figure 4 but for *MAE*.**

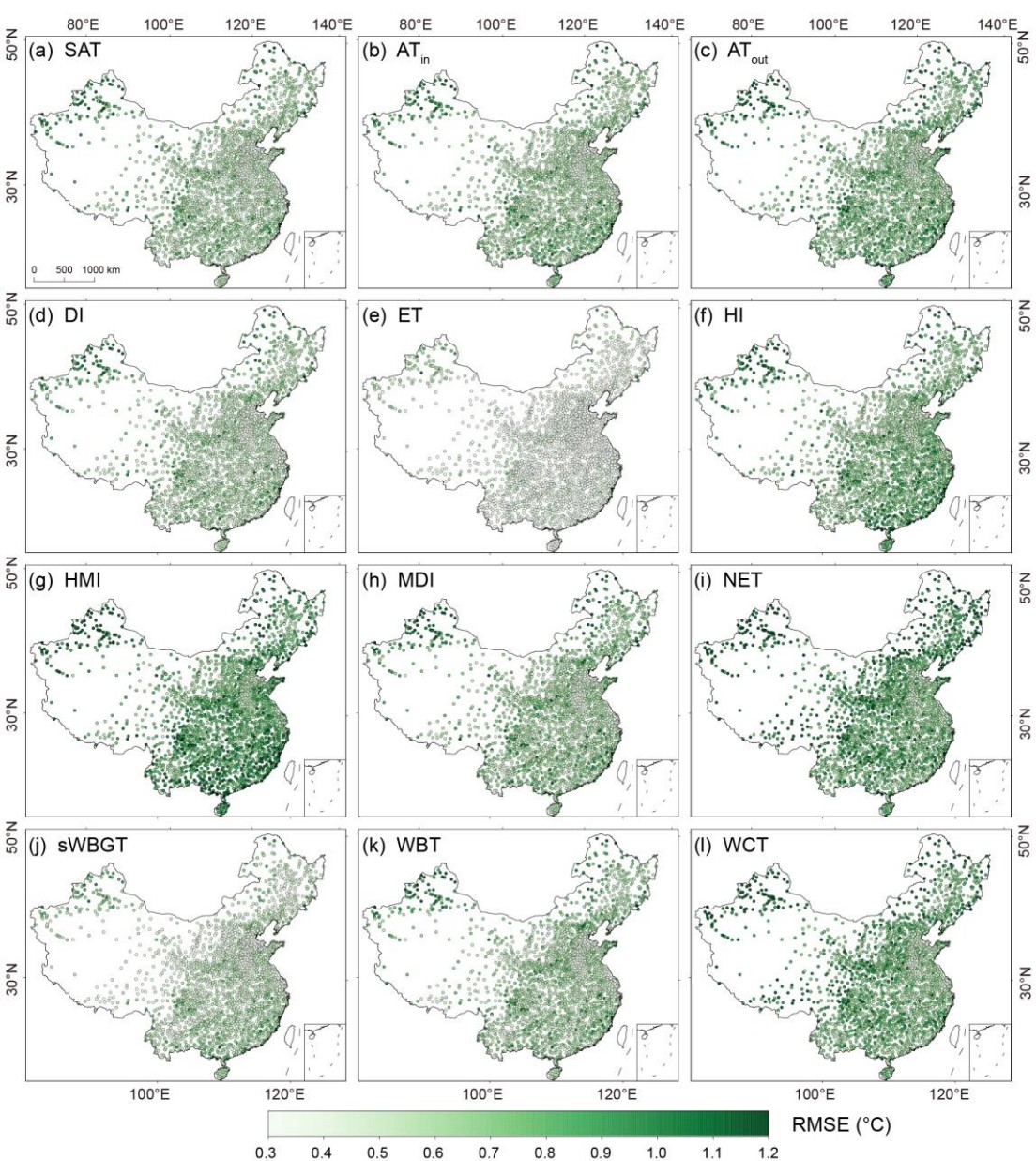


**Figure 6. As Figure 4 but for *RMSE*.**

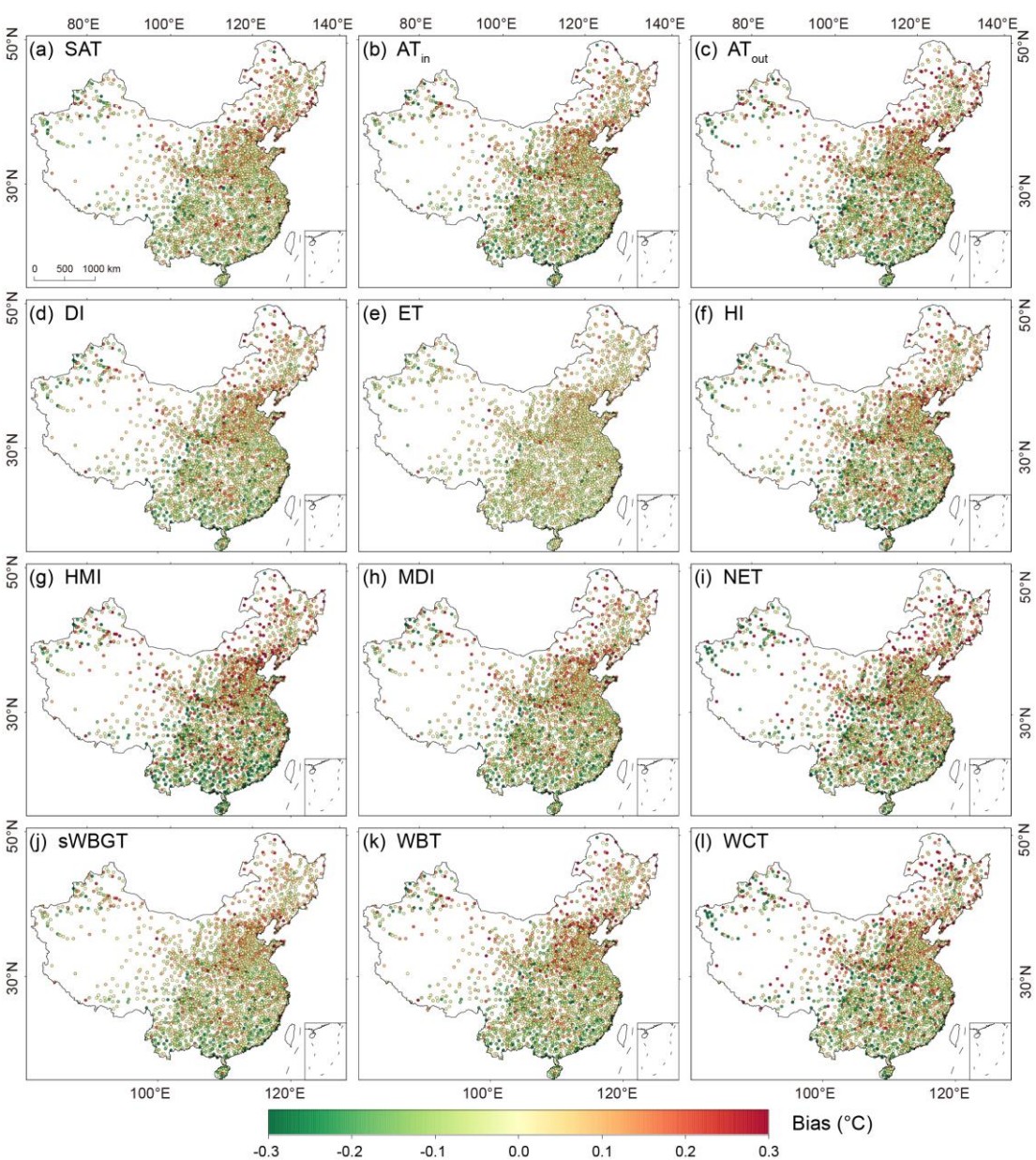


**Figure 7. As Figure 4 but for *Bias*.**


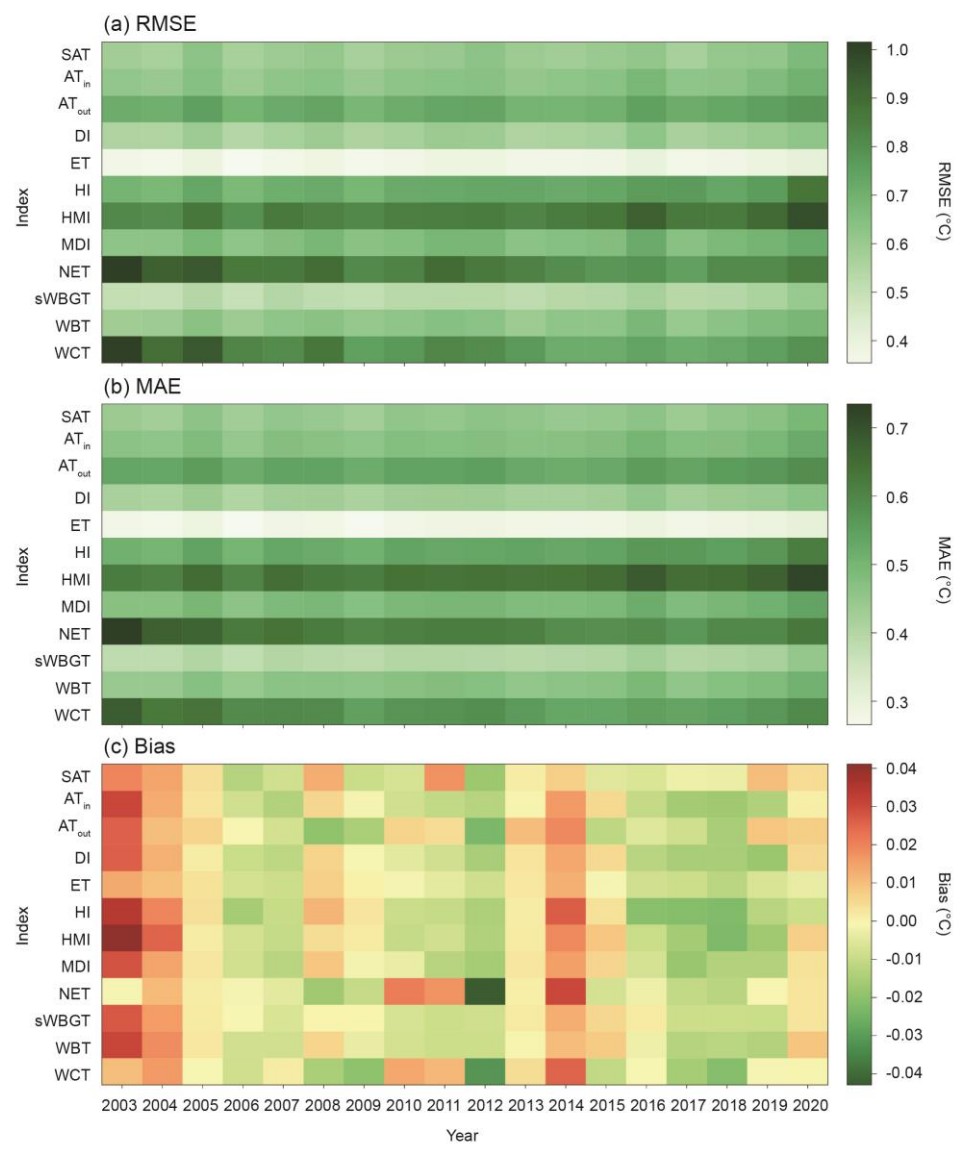


**Figure 8. Annual prediction accuracies of the 12 human thermal indices over the mainland of China during**
**2003–2020: (a)** *RMSE***, (b)** *MAE***, (c)** *Bias***.**

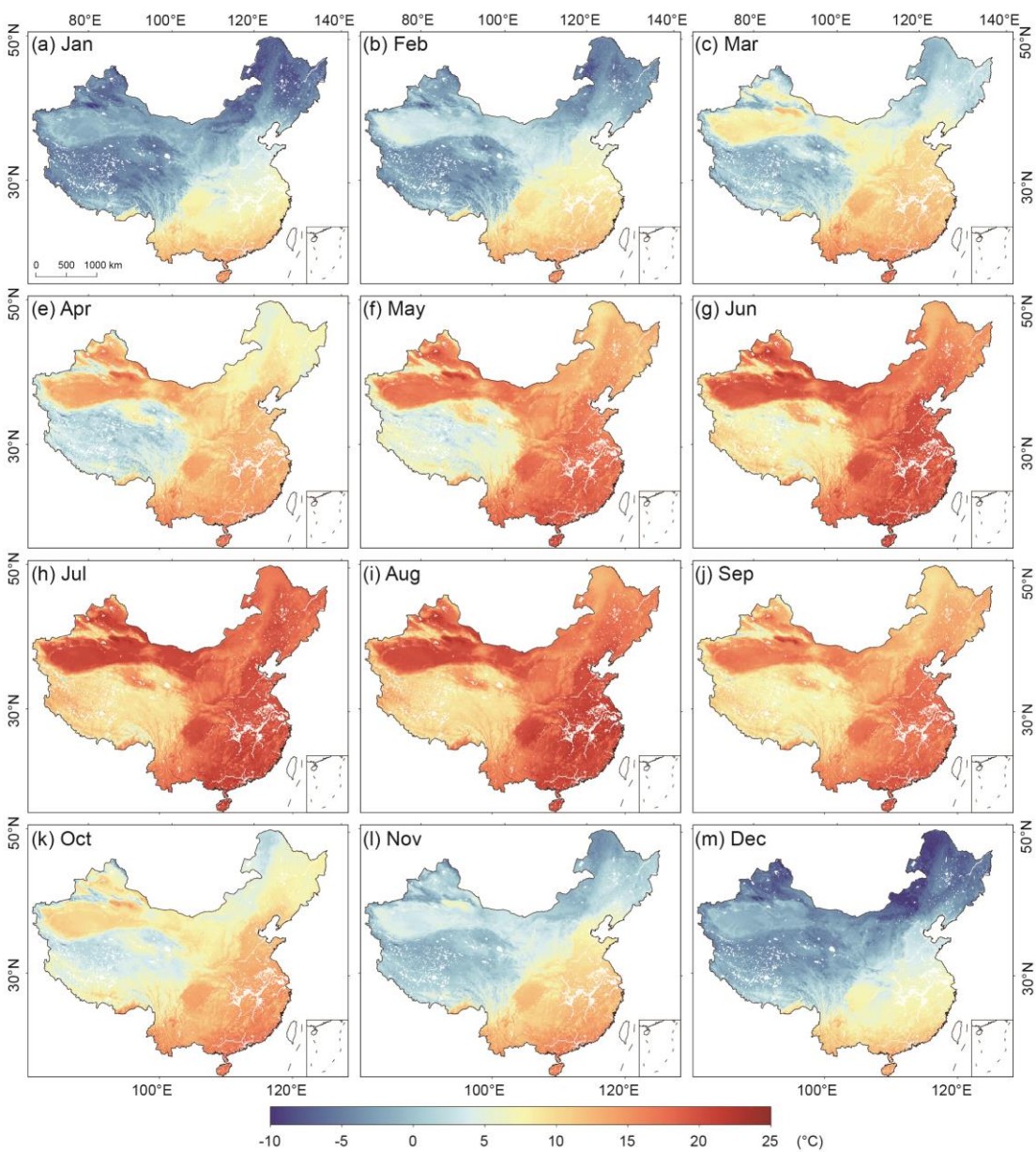


**Figure 9. Spatial distributions of the monthly mean ET over the mainland of China in 2020.**

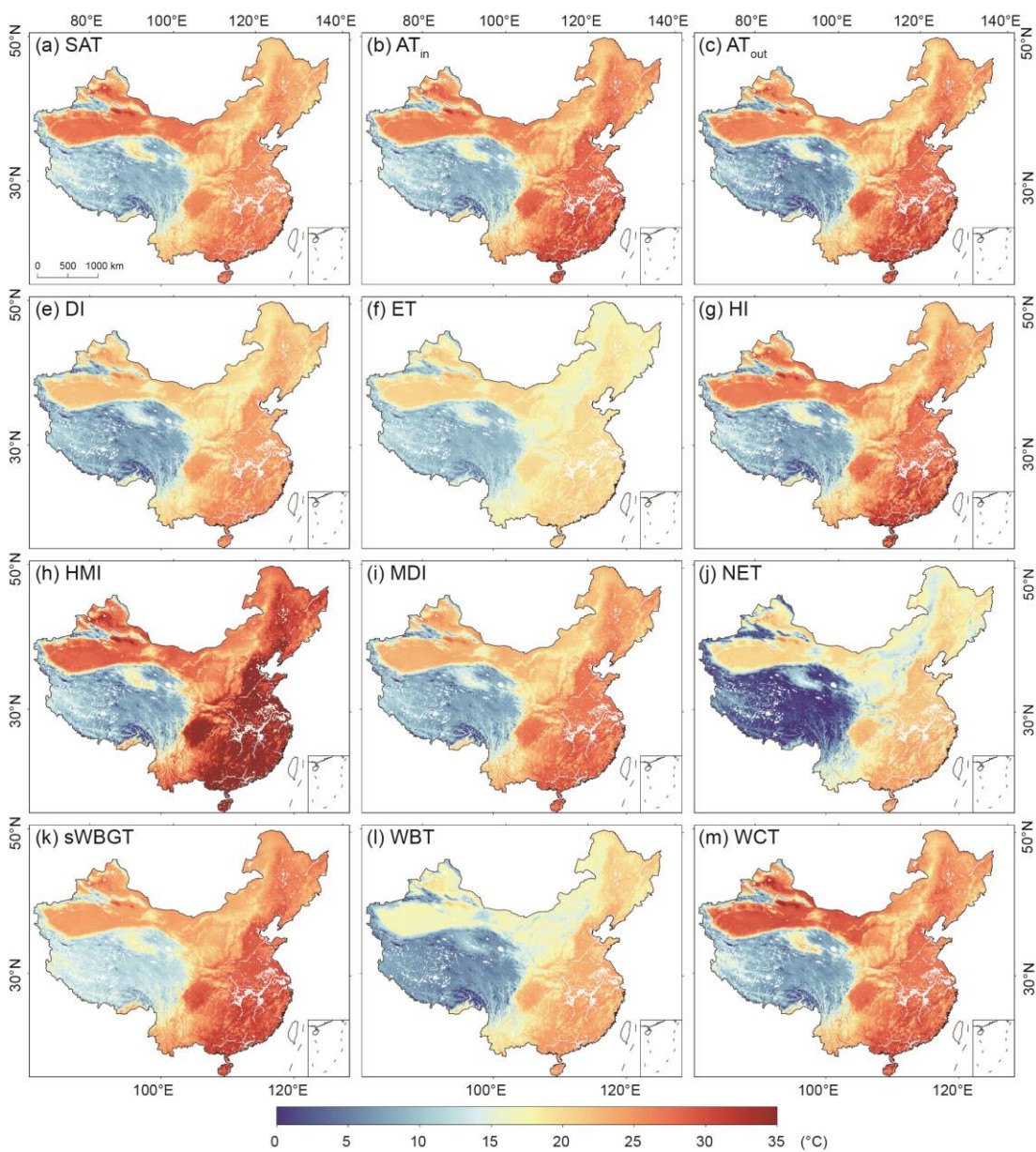


**Figure 10. Spatial distributions of the 12 human thermal indices over the mainland of China in July 2020.**


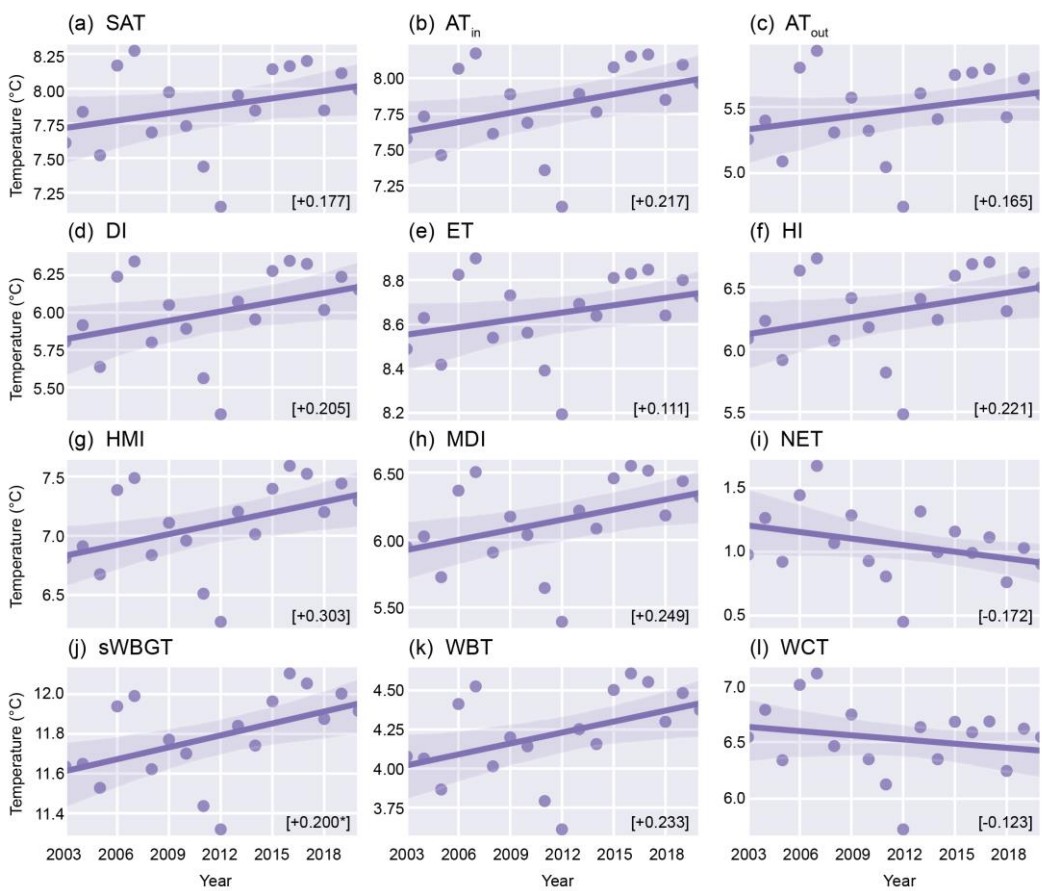


**Figure 11. Temporal changes of the 12 annually-averaged human thermal indices over the mainland of China**
**during 2003–2020. The line illustrates the linear trend, the number in the square bracket means the**
**corresponding trend per decade, and the asterisk next to the number indicates that the trends are significant**
**at the 0.05 level.**

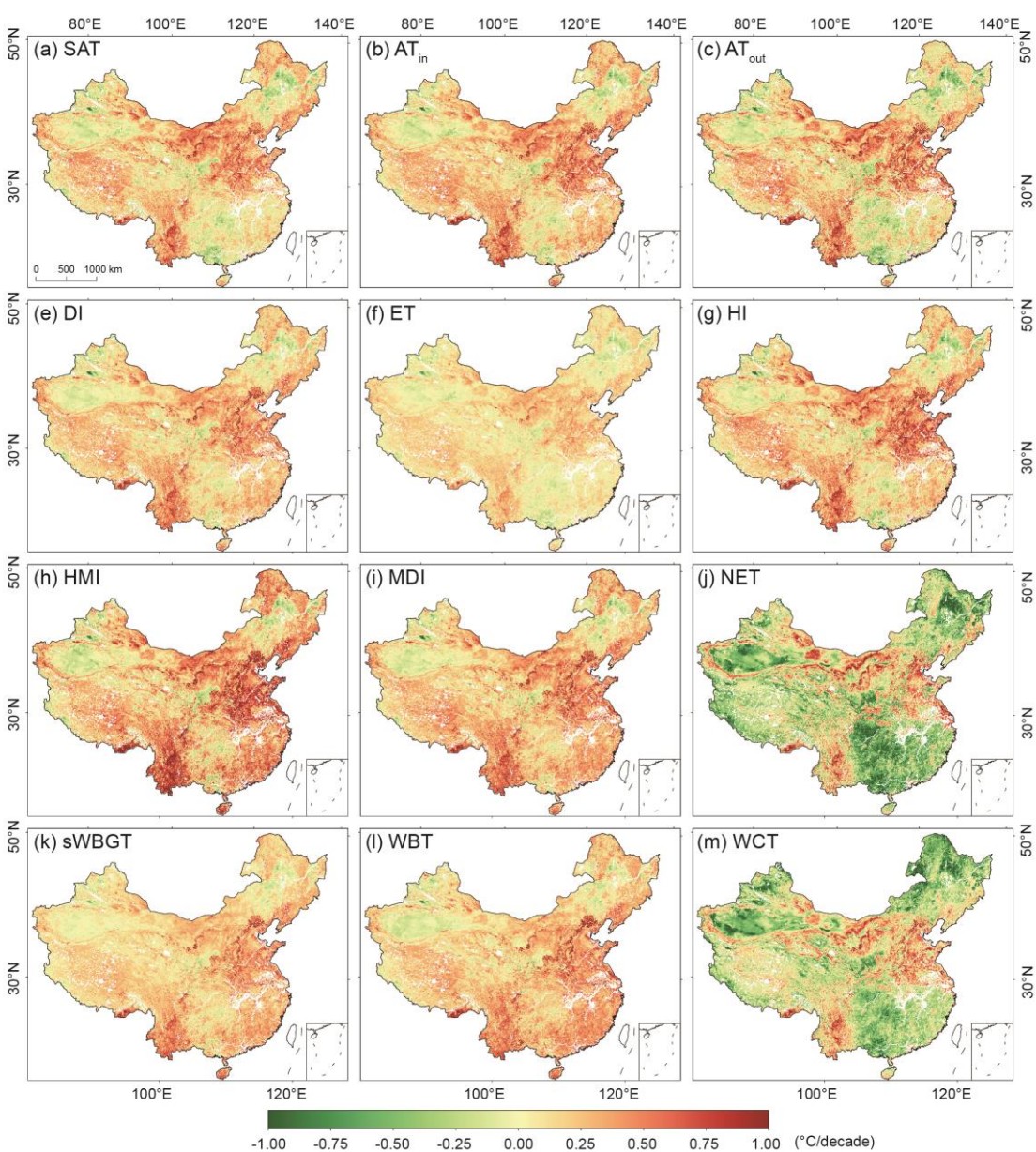


**Figure 12. Spatial distributions of the linear trends (unit: °C per decade) in the 12 annually-averaged human thermal indices over the mainland of China during 2003–2020.**


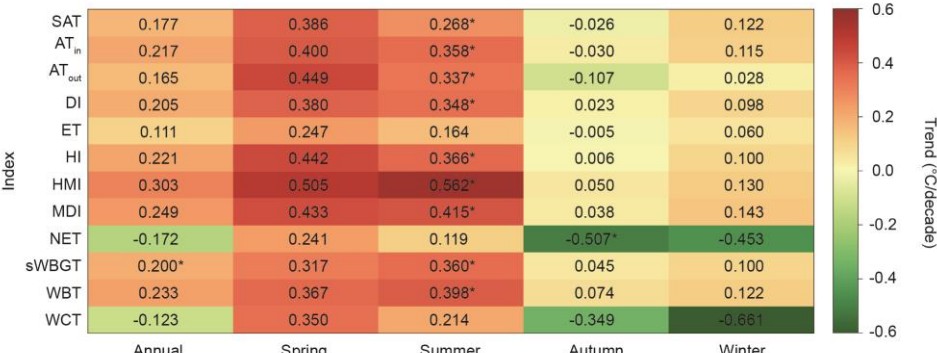


**Figure 13. Temporal trends of the 12 annually- and seasonally-averaged human thermal indices over the**
**mainland of China during 2003–2020. The number means linear trend per decade. The asterisk indicates that**
**the trends are significant at the 0.05 level.**

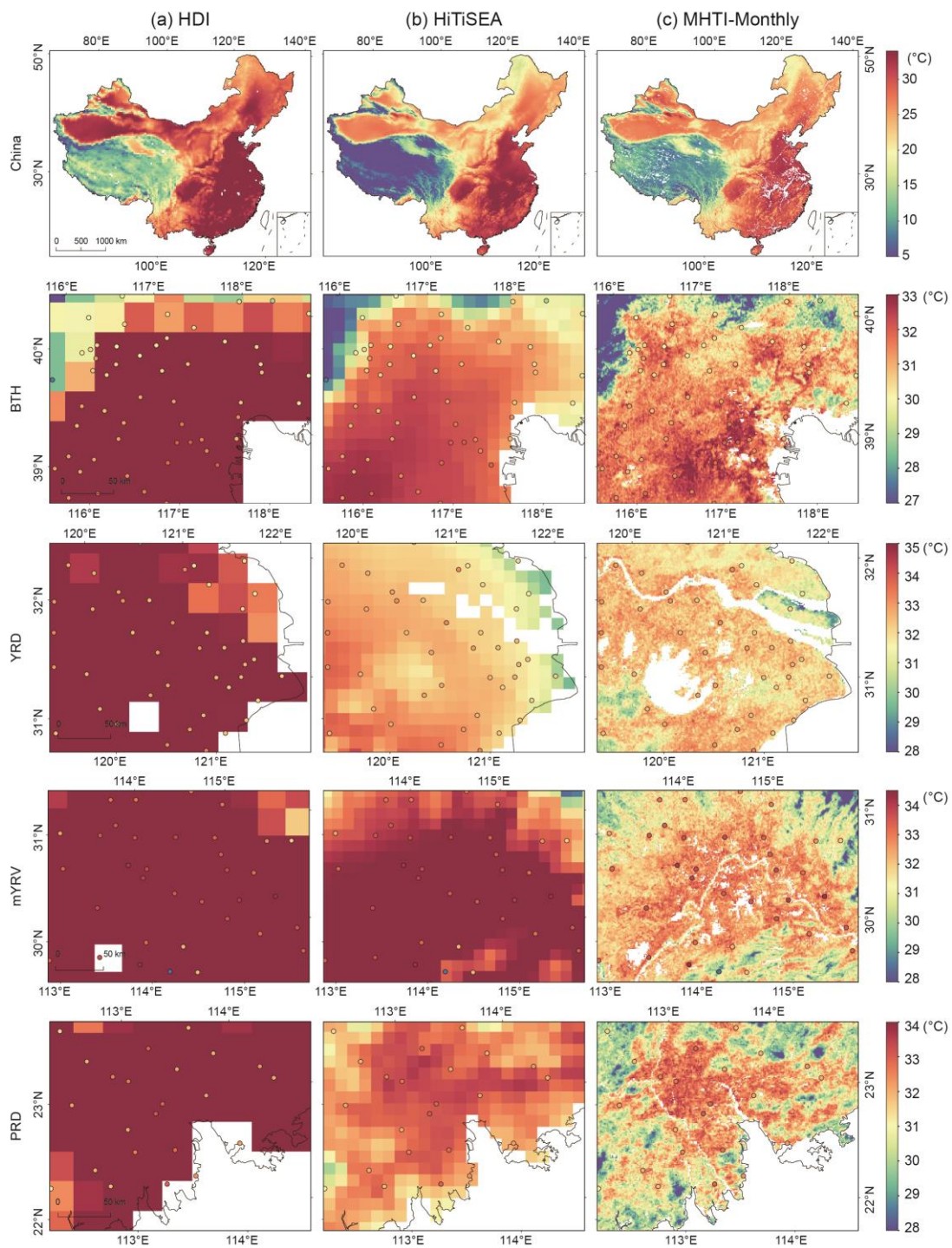


**Figure 14. Comparison of the spatial patterns among HDI_0p25_1970_2018 (HDI), HiTiSEA, and HiTIC-**
**Monthly for AT$_{in}$ over the mainland of China and its four largest UAs in July 2018: Beijing-Tianjin-Hebei**
**(BTH), Yangtze River Delta (YRD), middle Yangtze River Valley (mYRV) and Pearl River Delta (PRD).**
**Colored circles indicate the observed AT$_{in}$ values at individual meteorological stations.**

**Tables**
**Table 1. Grided datasets used in this study.**

| Category | Dataset | Spatial Resolution | Temporal Resolution | Variables | Data Source |
|----------|---------|-------------------|--------------------|-----------|-------------|
| Land surface temperature | A global seamless 1 km resolution daily land surface temperature dataset (2003-2020) | 1 km | Daily | Land surface temperature | Zhang et al. (2022b) |
| Land cover | MCD12Q1.006 | 500 m | Annual | Land cover classes in 1 km grids | Sulla-Menashe and Friedl (2019) |
| Elevation | MERIT DEM: Multi-Error-Removed Improved-Terrain DEM | 90 m | / | Aggregated elevation and slope in 1 km grids | Yamazaki et al. (2017) |
| Impervious surface | Tsinghua/FROM-GLC/GAIA/v10 | 30 m | Annual | Proportion of impervious surface in 1 km grids | Gong et al. (2020) |
| Population density | WorldPop | 1 km | Annual | Population density | Gaughan et al. (2013) |
| Temporal variation | / | / | / | Year, Month | / |


**Table 2. Equations of the human thermal indices for each station.**

| Abbreviation | Human thermal index | Computation model | Reference |
|---|---|---|---|
| $AT_{in}$ | Apparent Temperature (indoors) | $AT_{in} = -1.3 + 0.92 \times SAT + 2.2 \times E_a$ | Steadman (1979) |
| $AT_{out}$ | Apparent Temperature (outdoors, in the shade) | $AT_{out} = -2.7 + 1.04 \times SAT + 2 \times E_a - 0.65 \times V$ | Steadman (1984) |
| DI | Discomfort Index | $DI = 0.5 \times WBT + 0.5 \times SAT$ | Sohar et al. (1963) |
| ET | Effective Temperature | $ET = SAT - 0.4 \times (SAT - 10) \times (1 - 0.001 * RH)$ | Gagge et al. (1972) |
| HI | Heat Index* | $\begin{aligned} HI^* = &-8.784695 + 1.61139411 \times SAT - 2.338549 \times RH \\ &- 0.14611605 \times SAT \times RH \\ &- 1.2308094 \times 10^{-2} \times SAT^2 \\ &- 1.6424828 \times 10^{-2} \times RH^2 \\ &+ 2.211732 \times 10^{-3} \times SAT^2 \times RH \\ &+ 7.2546 \times 10^{-4} \times SAT \times RH^2 \\ &+ 3.582 \times 10^{-6} \times SAT^2 \times RH^2 \end{aligned}$ | Rothfusz and Headquarters (1990) |
| HMI | Humidex | $HMI = SAT + 0.5555 \times (0.1 \times E_a - 10)$ | Masterton et al. (1979) |
| MDI | Modified discomfort index | $MDI = 0.75 \times WBT + 0.38 \times SAT$ | Moran et al. (1998) |
| NET | Net Effective Temperature | $NET = 37 - \dfrac{37 - SAT}{0.68 - 0.0014 \times RH + \dfrac{1}{1.76 + 1.4 \times V^{0.75}}} - 0.29 \times SAT \times (1 - 0.01 \times RH)$ | Houghton and Yaglou (1923) |
| sWBGT | simplified Wet Bulb Globe Temperature | $sWBGT = 0.567 \times SAT + 0.0393 \times E_a + 3.94$ | Gagge and Nishi (1976) |
| WBT | Wet-bulb Temperature | $\begin{aligned} WBT = &SAT \times atan(0.151977 \times (RH + 8.313659)^{0.5}) \\ &+ atan(T + RH) - atan(RH - 1.676331) \\ &+ 0.00391838 \times RH^{1.5} \\ &\times atan(0.02301 \times RH) - 4.686035 \end{aligned}$ | Stull (2011) |
| WCT | Wind Chill Temperature | $\begin{aligned} WCT = &13.12 + 0.6215 \times SAT - 11.37 \times (V \times 3.6)^{0.16} \\ &+ 0.3965 \times SAT \times (V \times 3.6)^{0.16} \end{aligned}$ | Osczevski and Bluestein (2005) |

SAT is observed air temperature (°C), RH is relative humidity (%), V is wind speed (m/s), and $E_a$ is
actual water vapor pressure (kPa). Asterisk means that an adjustment is needed. All units of human
thermal indices in this study are in degrees Celsius (°C).

**Table 3. Overall prediction accuracies of the 12 human thermal indices over the mainland of China during**
**2003–2020.**

| Indices | $R^2$ | RMSE (°C) | MAE (°C) | Bias (°C) |
|---------|-------|-----------|----------|-----------|
| SAT | 0.9969 | 0.603 | 0.451 | -0.001 |
| $AT_{in}$ | 0.9971 | 0.635 | 0.478 | 0.002 |
| $AT_{out}$ | 0.9969 | 0.724 | 0.544 | 0.000 |
| DI | 0.9971 | 0.579 | 0.429 | 0.002 |
| ET | 0.9970 | 0.377 | 0.281 | 0.001 |
| HI | 0.9966 | 0.733 | 0.541 | 0.002 |
| HMI | 0.9968 | 0.859 | 0.645 | 0.000 |
| MDI | 0.9969 | 0.664 | 0.493 | 0.002 |
| NET | 0.9949 | 0.856 | 0.620 | 0.001 |
| sWBGT | 0.9967 | 0.535 | 0.401 | -0.001 |
| WBT | 0.9964 | 0.629 | 0.469 | 0.000 |
| WCT | 0.9959 | 0.807 | 0.579 | 0.002 |


**Table 4. Comparisons of the four thermal index datasets.**

| | ERA5-HEAT | HDI | HiTiSEA | HiTIC-Monthly |
|---|---|---|---|---|
| Spatial Resolution | 0.25°×0.25° | 0.25°×0.25° | 0.1°×0.1° | 1 km×1 km |
| Temporal Resolution | Hourly | Daily | Daily | Monthly |
| Spatial Coverage | Global | Global | South and East Asia | Mainland of China |
| Period | 1979–present | 1970–2018 | 1981–2019 | 2003–2020 |
| Thermal Indices | Mean Radiant Temperature (MRT), Universal Thermal Climate Index (UTCI) | Apparent Temperature indoors (ATind), two variants of Apparent Temperature outdoors in shade (ATot), Heat Index (HI), Humidex (HDEX), Wet Bulb Temperature (WBT), two variants of Wet Bulb Globe Temperature (WBGT), Thom Discomfort Index (DI), Windchill Temperature (WCT) | UTCI, indoor UTCI, outdoor shaded UTCI, MRT, Environment Stress Index (ESI), HI, Humidex, WBGT, WBT, WCT, AT, NET | SAT, $AT_{in}$, $AT_{out}$, DI, ET, HI, HMI, MDI, NET, sWBGT, WBT, WCT |