# Peer review of "HiTIC-Monthly: A Monthly High Spatial Resolution (1"

_Earth System Science Data, 2022_

## Author Comment (AC1)

**Responses to Review Comments**

Dear Editor,

Thank you and both reviewers for their careful review and valuable comments. We have carefully addressed all of the comments and revised our manuscript accordingly. Below please find our item-by-item responses. Please note that the review comments are in black, our responses are highlighted in blue, the extracts from the manuscript are in red, and **the new texts that have been added or changed** are in bold. Thank you very much for your time.

Best regards,

Ming Luo and Yongquan Zhao, on behalf of all authors

**#Reviewer 1**

Major comments.

The dataset is very import to study the climate change during this period over this region. The high spatial resolution and the monthly temporal resolution are basically reasonable, based on the previous datasets. The use of homogenous meteorological observation data is the right way to evaluate. This research is novel and belongs to the application of big data for large volume of Satellite data used, and machine learning algorithm. The authors are from 8 universities/colleges. The results supply important reference for other research fields like the field of geoscience, biology, sociology, and medicine and engineering.

*Response*: Many thanks for your appreciation of our work and the constructive comments. We have carefully revised our manuscript according to your comments and gave the following responses for your further review.

1) Line 415, harm, it is suggested to change the word. All 11 indices are calculated based on SAT, so the accuracy of this dataset is up to the accuracy of the dataset (Zhang 2022b) and the algorithm. SAT has important influence to the other 11 indices. How to understand these?

*Response*: Thank you for your comment. We have changed "harm" to "influence". The 11 human thermal indices were calculated from SAT at a daily scale (see Table 2 in the main text) for each station. Then, the daily thermal indices were monthly averaged and used for training together with the other predictors (e.g., LST, land cover, topography, and temporal variation) to estimate the spatial distribution of the 11 monthly indices at the 1 km grids. Ideally, the spatial prediction should be conducted based on monthly or even daily SAT. However, the SAT observations with full spatial coverage are not available, so we used monthly LST as an alternative to predict the spatial distribution of the monthly human thermal indices. In this case, the accuracy of monthly LST may influence the accuracy of our HiTIC-monthly dataset. We thereby

performed multiple accuracy assessments at various spatial (e.g., individual stations, the mainland of China, and its urban agglomerations) and temporal scales (e.g., monthly, yearly, and overall). The assessment results indicate that our dataset shows a desirable performance and exhibits good agreement with the observations in both spatial and temporal dimensions (see Section 4.1 of the main text), demonstrating the broad applicability of our dataset.

2) (Zhang 2022b) SAT is LST, but not Tair (1.5 meters above the surface). Why use LST but not Tair in Table 1 to compute the other 11 indices?

*Response*: As shown in our response to your comment #1, we first computed daily thermal indices observed at the weather stations based on daily SAT and other meteorological observations (see Table 2), and then aggregated them to monthly values. These spatially-discrete (station-based) monthly indices were then used for predicting spatially-continuous human thermal indices on 1 km grids, which adopted seamless monthly LST (seamless SAT is not available) and other related variables as predictors. Previous studies have shown that LST is highly correlated with SAT and can be used as a proxy to predict SAT and SAT-related human thermal indices (Zhu et al., 2013; Zhu et al., 2019; Shamir and Georgakakos, 2014). Our results also demonstrate that using LST can generate a desirable accuracy in predicting monthly human thermal indices.

3) Line 157-158 "Section 6 compares our products with two existing datasets, and the main findings of this paper are summarized in Section 7". Where is section 6 and section 7? They are data availability and conclusions.

*Response*: Thank you for pointing out this issue. We have revised the description as (Lines 154–155): **"Section 5 compares our products with two existing datasets. Section 6 provides data availability, and the main findings of this paper are summarized in Section 7."**

4) Why not keep Figures S1-S9, Tables S1-S5 as formal ones? Please consider again which to keep in the manuscript.

*Response*: Thanks a lot for your advice. We have moved Tables S1–S2 to the main text in the current revision, and we decided to keep Figures S1–S9 and Tables S3–S5 in the supplementary material because there are already 14 figures and 4 tables in the main text. Including more figures and tables may not meet the requirement of the journal of ESSD and influence the readability of our manuscript.

5) How to get monthly data from MODIS daily mid-daytime 13:30 and mid-nighttime 01:30 LST? The monthly mean is different considering of the diurnal variation and satellite observations from ascending orbits and descending orbits.

*Response*: Thank you for the comment. Our monthly LST values were calculated by averaging daily LST in the corresponding month in the calendar years of 2003−2020. The daily mean LST values were obtained by averaging four MODIS observations in a day, which correspond to mid-daytime and mid-nighttime observations from ascending and descending orbits. In the current revised version, we have added the following explanations (Lines 175–179):

**"We used monthly LST as one of the inputs to predict the spatial distribution of 12 thermal indices. Monthly LST values were calculated by averaging daily LST, which was obtained by averaging four observations in a day, including mid-daytime and mid-nighttime observations from ascending and descending orbits of MOD11A1 (Terra) and MYD11A1 (Aqua). More details about the LST data are described in Zhang et al. (2022b)."**

6) Line 190-191, how to compute and use the covariates?

*Response*: Thank you for the question. The covariates directly come from the data providers or producers, and these information is listed in Table 1. All these covariates were pre-processed to have the same spatial extent, projection, and spatial resolution for predicting human thermal indices with full spatial coverage at the 1 km resolution.

7) HiTIC? What is i? Is it HTI (human thermal index)? No "I" after "human" and before " thermal".

*Response*: "HiTIC" stands for **Hi**gh spatial resolution **T**hermal **I**ndex **C**ollection. We have changed the title to "**HiTIC-Monthly: A Monthly High Spatial Resolution (1 km) Thermal Index Collection over China during 2003−2020**" to avoid misunderstanding.

Minor comments.

1) Add abbr. of LGBM when it is first used. Light Gradient Boosting Machine. Check others, please.

*Response*: Thank you for your suggestion. The full name of LGBM has been added (Lines 36–37): "In this collection, 12 commonly-used thermal indices were generated by **the Light Gradient Boosting Machine (LGBM)** learning algorithm from multi-source gridded data."

2) Author contribution. It seems only H.Z is responsible for analysis and data processing, others are all involved in writing. For computations, are there anyone else in the author list?

*Response*: The first author H.Z. and two corresponding authors of M.L. and Y.Z. are responsible for data analysis and computation. We also wish to clarify that this work involves a large number of tasks (including but not limited to the selection of predictors, the accuracy assessment, the discussion and interpretation of the results, and the

comparison with other datasets), and all other authors in the list contributed to discussion, investigation, and writing.

3) Line 439. The unit of the dataset is degree or 0.01 degree? Is 0.01 the scale factor?

*Response*: Yes, the unit of the dataset is 0.01 degree Celsius (°C). The values are stored in integer type (Int16) for saving storage and need to be multiplied by 100 when in use. The following updates have been made in the main text (Lines 452–454): "The unit of the dataset is 0.01 degree Celsius (°C)**, and the values are stored in an integer type (Int16) for saving storage space, and need to be multiplied by 100 when in use.**"

4) Please revise the followings.

- Line 247-248, as the figure displays

*Response*: Revised.

- Line 384, Figures S8j&m. "Figure S8", not figures, and add a blank before j.

*Response*: Revised.

It is suggested to delete equations (1-2) and (3-6), add the reference.

*Response*: Thanks for the suggestion. With respect, we believe that these equations are helpful for readers to understand the details more easily, and thus can be kept in the main text. As suggested, we have added the relevant references in the current revision:

Lines 202–203: "$E_a$ is derived from T and RH rather than directly observed at meteorological stations (Eqs. 1~2; **Bolton (1980)**)."

Lines 235–236: "Four statistic metrics, namely, determination coefficient ($R^2$), Mean Absolute Error (MAE), RMSE, and Bias **(Rice, 2006)**."

- If necessary, please keep the same description of the time period for there are many kinds in the manuscript, like 2003~2020, from 2003 to 2020, during the year

2003~2020, (2003 to 2020), during 2003~2020, from January 2003 to December 2020, 2003-2020.

*Response*: Thanks for your suggestion. We have changed all descriptions of the time period to "**during 2003−2020**".

- Line 755. Delete the last sentence. Line 759, change : to . after 2003~2020.

*Response*: Revised.

- Add longitude and latitude (or the numerical scope signs) in Fig 1, 4-7, 9-12, 14.

*Response*: Added.

Figure 8. Prediction accuracies of 12 human thermal indices… Add 12. In individual years, change the description. Time series? Add the description of (a) (b) (c).

*Response*: Added.

- Figure 10. Spatial distributions of 12 human thermal indices... Add 12.

*Response*: Added.

- Line 780. Figure 11. National average, are you sure to write like this? Check it in other places in the manuscript. Just use average is fine. Delete "straight". Line 789, Figure 13, national, delete this. Add 12, 12 human thermal indices.

*Response*: Thank you. We have revised these sentences accordingly. For example, we have revised the caption of Figure 11 as (Lines 814–815): "Temporal changes of **the 12 annually-averaged human thermal indices over the mainland of China during 2003−2020**."

- Line 785, Figure 12, add 12. "the trends of annual mean", please consider how to express better. Inter-annual variation?

*Response*: Thank you for the comment. The sentence has been rewritten as (Lines 820– 821): "Spatial distributions of the **linear** trends (unit: °C per decade) in **the 12**

**annually-averaged human thermal indices** over the mainland of China **during 2003–2020**."

- Line 795, Figure 14. HITIC, is it "i"? in mainland China, is it over mainland of China? Please check the description of "mainland China" in the manuscript. In July 2018, move to the end of four major UAs. Delete i.e., change to :.

*Response*: Thank you for the comment. We have revised these sentences as follows (Lines 829-832):

"Comparison of the spatial patterns among HDI_0p25_1970_2018 (HDI), HiTiSEA, and **HiTIC-Monthly** for ATin **over the mainland of China and its four largest UAs** in July 2018: Beijing-Tianjin-Hebei (BTH), Yangtze River Delta (YRD), middle Yangtze River Valley (mYRV) and Pearl River Delta (PRD). Colored circles indicate the observed ATin values at individual meteorological stations."

**#Reviewer 2**

The authors have produced a high-resolution (1 km×1 km) thermal index collection at a monthly scale (HiTIC-Monthly) in China during 2003 to 2020, with 12 widely used human thermal indices. The authors have created a high-resolution products for quantifying thermal index in China, which is valuable to the scientific community. I have some comments to be addressed by the authors.

Response: Thank you very much for your helpful comments and suggestions. We have carefully revised our manuscript based on your comments. Below please find our item-by-item responses.

1. The biggest concern is the temporal resolution. Why do the authors choose monthly resolution, rather than daily? Daily products would be extremely useful to characterize extreme events, which are of societal importance.

*Response*: Thank you very much for your comment. We agree with you that daily products could be useful to characterize extreme events. However, a daily thermal indices dataset with full spatial coverage requires daily covariates that have full spatial coverage as well, and these data are not available. We produced a long-term monthly thermal indices dataset with high accuracies. The monthly dataset could be useful for the studies of climate change and urban climate and environment across China and has the potential for investigations of heat-related illnesses and deaths. A daily high-resolution human thermal index collection (HiTIC-Daily) will be produced and released in our future studies.

2. Lines 168-169: How about the impacts of precipitation on thermal indices? Have the authors considered precipitation as a covariate?

*Response*: Thank you for bringing our attention to precipitation. We did not include precipitation as a covariate because the precipitation data are not normally distributed.

More importantly, they exhibit many zero values in many regions of China (especially in the dry season), which would increase the uncertainty of the spatial prediction.

3. Figures 7 and 8: The results indicate spatial variability of bias in the thermal indices. What factors drive the spatial variability of the bias? Meanwhile, there is also temporal variability in the bias (Figure 8), and what is the drivers of this variability? Are the spatial and temporal variabilities of the bias related to background climates?

*Response*: Thank you for the comment. As shown in Figure 7, the bias values range from -0.3 °C to +0.3 °C. It exhibits a zonal variation, i.e., positive values tend to distribute in northern China, and negative values are mainly located in the south. This variation is likely caused by the lower temperature in the north and higher temperature in the south. This issue of overestimation of low values and underestimation of high values is a common problem in machine learning prediction (Wu et al., 2022; Li et al., 2020; Uddin et al., 2022; Cho et al., 2020), but the overall estimations in our study are still reliable (see the evaluation results in Section 4.1). A similar situation can also be seen in the temporal variability of the bias (Figure 8). Positive bias values are more likely seen in early periods with lower temperatures, and negative bias values tend to appear in more recent periods with higher temperatures.

4. One way of evaluating the quality of these products is to evaluate the EOFs of these products. For example, what are the first three EOFs in each product? How do the temporal coefficients change over time across these products? Such spatial-temporal evaluation would be desirable.

*Response*: Thank you very much for this suggestion. We have followed your suggestion and evaluated the first three EOF modes of the 12 thermal indices. The spatial patterns and corresponding temporal coefficients of these EOFs are depicted in Figures R1−R4 (see also Figure S10−S13 in the Supporting Information). As Figure R1 shows, the leading EOF (EOF1) of all 12 thermal indices exhibit highly consistent spatial

distribution with higher values in the north and lower values in the south. Their temporal variations are also similar to each other (Figure R2). The second and third EOF modes (EOF2 and EOF3) are also similar among different thermal indices (except EOF3 of NET, Figures R2−R4). These results demonstrate the desirable quality of our products. We have thus included the following discussions in the revised manuscript (Lines 332-338):

"**The dominant modes of these indices are further examined by applying the empirical orthogonal function (EOF) analysis (Figures S10−S13). As Figure S10 shows, the leading EOF (EOF1) of all 12 thermal indices exhibit highly consistent spatial distribution with higher values in the northern region and lower values in the south. Their temporal variations are also similar to each other (Figure S11). The second and third EOF modes (EOF2 and EOF3) are also similar among different thermal indices (except EOF3 of NET, Figures S11−S13). These results demonstrate the desirable quality of our products.**"

[Figure]

**Figure R1. Spatial distributions of the leading empirical orthogonal function (EOF1) of the 12 human thermal indices over the mainland of China. The numbers in square brackets represent the percentage of the variance explained by the corresponding EOF mode.**

[Figure]

**Figure R2. Time series of the principal components (PCs) corresponding to the first three EOF modes of the 12 human thermal indices over the mainland of China during 2003–2020.**

[Figure]

**Figure R3. As Figure R1 but for the second EOF (EOF2).**

[Figure]

**Figure R4. As Figure R1 but for the third EOF (EOF3).**

**References**

Cho, D., Yoo, C., Im, J., and Cha, D. H.: Comparative Assessment of Various Machine Learning‑Based Bias Correction Methods for Numerical Weather Prediction Model Forecasts of Extreme Air Temperatures in Urban Areas, Earth and Space Science, 7, 10.1029/2019ea000740, 2020.

Li, Y., Li, M., Li, C., and Liu, Z.: Forest aboveground biomass estimation using Landsat 8 and Sentinel-1A data with machine learning algorithms, Sci Rep, 10, 9952, 10.1038/s41598-020-67024-3, 2020.

Shamir, E. and Georgakakos, K. P.: MODIS Land Surface Temperature as an index of surface air temperature for operational snowpack estimation, Remote Sensing of Environment, 152, 83-98, https://doi.org/10.1016/j.rse.2014.06.001, 2014.

Uddin, M. G., Nash, S., Mahammad Diganta, M. T., Rahman, A., and Olbert, A. I.: Robust machine learning algorithms for predicting coastal water quality index, J Environ Manage, 321, 115923, 10.1016/j.jenvman.2022.115923, 2022.

Wu, J., Fang, H., Qin, W., Wang, L., Song, Y., Su, X., and Zhang, Y.: Constructing High-Resolution (10 km) Daily Diffuse Solar Radiation Dataset across China during 1982–2020 through Ensemble Model, Remote Sensing, 14, https://doi.org/10.3390/rs14153695, 2022.

Zhu, W., Lű, A., and Jia, S.: Estimation of daily maximum and minimum air temperature using MODIS land surface temperature products, Remote Sensing of Environment, 130, 62-73, https://doi.org/10.1016/j.rse.2012.10.034, 2013.

Zhu, X., Zhang, Q., Xu, C. Y., Sun, P., and Hu, P.: Reconstruction of high spatial resolution surface air temperature data across China: A new geo-intelligent multisource data-based machine learning technique, Sci Total Environ, 665, 300-313, https://doi.org/10.1016/j.scitotenv.2019.02.077, 2019.

---

## Author Comment (AC3)

**Responses to Review Comments**

Dear Editor,

Thank you and all the reviewers for their careful review and valuable comments. We have carefully addressed all of the comments and revised our manuscript accordingly. Below please find our item-by-item responses. Please note that the review comments are in black, our responses are highlighted in blue, the extracts from the manuscript are in red, and the new texts that have been added or changed are in bold. Thank you very much for your time.

Best regards,

Ming Luo and Yongquan Zhao, on behalf of all authors

**#Reviewer 1**

Major comments.

The dataset is very import to study the climate change during this period over this region. The high spatial resolution and the monthly temporal resolution are basically reasonable, based on the previous datasets. The use of homogenous meteorological observation data is the right way to evaluate. This research is novel and belongs to the application of big data for large volume of Satellite data used, and machine learning algorithm. The authors are from 8 universities/colleges. The results supply important reference for other research fields like the field of geoscience, biology, sociology, and medicine and engineering.

*Response*: Many thanks for your appreciation of our work and the constructive comments. We have carefully revised our manuscript according to your comments and gave the following responses for your further review.

1) Line 415, harm, it is suggested to change the word. All 11 indices are calculated based on SAT, so the accuracy of this dataset is up to the accuracy of the dataset (Zhang 2022b) and the algorithm. SAT has important influence to the other 11 indices. How to understand these?

*Response*: Thank you for your comment. We have changed "harm" to "influence". The 11 human thermal indices were calculated from SAT at a daily scale (see Table 2 in the main text) for each station. Then, the daily thermal indices were monthly averaged and used for training together with the other predictors (e.g., LST, land cover, topography, and temporal variation) to estimate the spatial distribution of the 11 monthly indices at the 1 km grids. Ideally, the spatial prediction should be conducted based on monthly or even daily SAT. However, the SAT observations with full spatial coverage are not available, so we used monthly LST as an alternative to predict the spatial distribution of the monthly human thermal indices. In this case, the accuracy of monthly LST may influence the accuracy of our HiTIC-monthly dataset. We thereby performed multiple accuracy assessments at various spatial (e.g., individual stations, the mainland of China, and its urban agglomerations) and temporal scales (e.g., monthly, yearly, and overall). The assessment results indicate that our dataset shows a desirable performance and exhibits good agreement with the observations in both spatial and temporal dimensions (see Section 4.1 of the main text), demonstrating the broad applicability of our dataset.

2) (Zhang 2022b) SAT is LST, but not Tair (1.5 meters above the surface). Why use LST but not Tair in Table 1 to compute the other 11 indices?

*Response*: As shown in our response to your comment #1, we first computed daily thermal indices observed at the weather stations based on daily SAT and other meteorological observations (see Table 2), and then aggregated them to monthly values. These spatially-discrete (station-based) monthly indices were then used for predicting spatially-continuous human thermal indices on 1 km grids, which adopted seamless monthly LST (seamless SAT is not available) and other related variables as predictors. Previous studies have shown that LST is highly correlated with SAT and can be used as a proxy to predict SAT and SAT-related human thermal indices (Shamir & Georgakakos, 2014; W. Zhu, Lǚ, & Jia, 2013; X. Zhu, Zhang, Xu, Sun, & Hu, 2019). Our results also demonstrate that using LST can generate a desirable accuracy in predicting monthly human thermal indices.

3) Line 157-158 "Section 6 compares our products with two existing datasets, and the main findings of this paper are summarized in Section 7". Where is section 6 and section 7? They are data availability and conclusions.

*Response*: Thank you for pointing out this issue. We have revised the description as (Lines 156–158): "**Comparisons on our products with two existing datasets are in Section 5, data availability is provided in Section 6. The main findings of this paper are summarized in Section 7.**"

4) Why not keep Figures S1-S9, Tables S1-S5 as formal ones? Please consider again which to keep in the manuscript.

*Response*: Thanks a lot for your advice. We have moved Tables S1–S2 to the main text in the current revision, and we decided to keep Figures S1–S9 and Tables S3–S5 in the supplementary material because there are already 14 figures and 4 tables in the main text. Including more figures and tables may not meet the requirement of the journal of ESSD and influence the readability of our manuscript.

5) How to get monthly data from MODIS daily mid-daytime 13:30 and mid-nighttime 01:30 LST? The monthly mean is different considering of the diurnal variation and satellite observations from ascending orbits and descending orbits.

*Response*: Thank you for the comment. Our monthly LST values were calculated by averaging daily LST in the corresponding month in the calendar years of 2003−2020. The daily mean LST values were obtained by averaging four MODIS observations in a day, which correspond to mid-daytime and mid-nighttime observations from ascending and descending orbits. In the current revised version, we have added the following explanations (Lines 177–182):

**"We used monthly LST as one of the inputs to predict the spatial distribution of 12 thermal indices. Monthly LST values were calculated by averaging daily LST, which was obtained by averaging four observations in a day, including mid-daytime and mid-nighttime observations from ascending and descending orbits of MOD11A1 (Terra) and MYD11A1 (Aqua). More details about the LST data are described in Zhang et al. (2022b)."**

6) Line 190-191, how to compute and use the covariates?

*Response*: Thank you for the question. The covariates directly come from the data providers or producers, and these information is listed in Table 1. All these covariates were pre-processed to have the same spatial extent, projection, and spatial resolution for predicting human thermal indices with full spatial coverage at the 1 km resolution.

7) HiTIC? What is i? Is it HTI (human thermal index)? No "I" after "human" and before " thermal".

 *Response*: "HiTIC" stands for **Hi**gh spatial resolution **T**hermal **I**ndex **C**ollection. We have changed the title to "**HiTIC-Monthly: A Monthly High Spatial Resolution (1 km) Human Thermal Index Collection over China during 2003−2020**" to avoid misunderstanding.

Minor comments.

1)  Add abbr. of LGBM when it is first used. Light Gradient Boosting Machine. Check others, please.

*Response*: Thank you for your suggestion. The full name of LGBM has been added (Lines 38–39): "In this collection, 12 commonly-used thermal indices were generated by **the Light Gradient Boosting Machine (LGBM)** learning algorithm from multi-source gridded data."

2)  Author contribution. It seems only H.Z is responsible for analysis and data processing, others are all involved in writing. For computations, are there anyone else in the author list?

*Response*: The first author H.Z. and two corresponding authors of M.L. and Y.Z. are responsible for data analysis and computation. We also wish to clarify that this work involves a large number of tasks (including but not limited to the selection of predictors, the accuracy assessment, the discussion and interpretation of the results, and the comparison with other datasets), and all other authors in the list contributed to discussion, investigation, and writing.

3) Line 439. The unit of the dataset is degree or 0.01 degree? Is 0.01 the scale factor?

*Response*: Yes, the unit of the dataset is 0.01 degree Celsius (°C). The values are stored in integer type (Int16) for saving storage and need to be multiplied by 100 when in use. The following updates have been made in the main text (Lines 454–456): "The unit of the dataset is 0.01 degree Celsius (°C)**, and the values are stored in an integer type (Int16) for saving storage space, and need to be divided by 100 to get the values in degree Celsius when in use.**"

4) Please revise the followings.

● Line 247-248, as the figure displays

*Response*: Revised.

● Line 384, Figures S8j&m. "Figure S8", not figures, and add a blank before j.

*Response*: Revised.

It is suggested to delete equations (1-2) and (3-6), add the reference.

*Response*: Thanks for the suggestion. With respect, we believe that these equations are helpful for readers to understand the details more easily, and thus can be kept in the main text. As suggested, we have added the relevant references in the current revision:

Lines 204–205: "$E_a$ is derived from T and RH rather than directly observed at meteorological stations (Eqs. 1~2; **Bolton (1980)**)."

Lines 237–238: "Four statistic metrics, namely, determination coefficient ($R^2$), Mean Absolute Error (MAE), RMSE, and Bias **(Rice, 2006)**."

● If necessary, please keep the same description of the time period for there are many kinds in the manuscript, like 2003~2020, from 2003 to 2020, during the year

2003~2020, (2003 to 2020), during 2003~2020, from January 2003 to December 2020, 2003-2020.

*Response*: Thanks for your suggestion. We have changed all descriptions of the time period to "**during 2003−2020**".

● Line 755. Delete the last sentence. Line 759, change : to . after 2003~2020.

*Response*: Revised.

● Add longitude and latitude (or the numerical scope signs) in Fig 1, 4-7, 9-12, 14.

*Response*: Added.

Figure 8. Prediction accuracies of 12 human thermal indices… Add 12. In individual years, change the description. Time series? Add the description of (a) (b) (c).

*Response*: Added.

● Figure 10. Spatial distributions of 12 human thermal indices... Add 12.

*Response*: Added.

● Line 780. Figure 11. National average, are you sure to write like this? Check it in other places in the manuscript. Just use average is fine. Delete "straight". Line 789, Figure 13, national, delete this. Add 12, 12 human thermal indices.

*Response*: Thank you. We have revised these sentences accordingly. For example, we have revised the caption of Figure 11 as (Lines 811–812): "Temporal changes of **the 12 annually-averaged human thermal indices over the mainland of China during 2003−2020**."

● Line 785, Figure 12, add 12. "the trends of annual mean", please consider how to express better. Inter-annual variation?

*Response*: Thank you for the comment. The sentence has been rewritten as (Lines 817–818): "Spatial distributions of the **linear** trends (unit: °C per decade) in **the 12**

**annually-averaged human thermal indices** over the mainland of China **during 2003−2020**."

- Line 795, Figure 14. HITIC, is it "i"? in mainland China, is it over mainland of China? Please check the description of "mainland China" in the manuscript. In July 2018, move to the end of four major UAs. Delete i.e., change to :.

*Response*: Thank you for the comment. We have revised these sentences as follows (Lines 826-829):

"Comparison of the spatial patterns among HDI_0p25_1970_2018 (HDI), HiTiSEA, and **HiTIC-Monthly** for $AT_{in}$ **over the mainland of China and its four largest UAs** in July 2018: Beijing-Tianjin-Hebei (BTH), Yangtze River Delta (YRD), middle Yangtze River Valley (mYRV) and Pearl River Delta (PRD). Colored circles indicate the observed $AT_{in}$ values at individual meteorological stations."

**#Reviewer 2**

The authors have produced a high-resolution (1 km×1 km) thermal index collection at a monthly scale (HiTIC-Monthly) in China during 2003 to 2020, with 12 widely used human thermal indices. The authors have created a high-resolution products for quantifying thermal index in China, which is valuable to the scientific community. I have some comments to be addressed by the authors.

Response: Thank you very much for your helpful comments and suggestions. We have carefully revised our manuscript based on your comments. Below please find our item-by-item responses.

1. The biggest concern is the temporal resolution. Why do the authors choose monthly resolution, rather than daily? Daily products would be extremely useful to characterize extreme events, which are of societal importance.

*Response*: Thank you very much for your comment. We agree with you that daily products could be useful to characterize extreme events. However, a daily thermal indices dataset with full spatial coverage requires daily covariates that have full spatial coverage as well, and these data are not available. We produced a long-term monthly thermal indices dataset with high accuracies. The monthly dataset could be useful for the studies of climate change and urban climate and environment across China and has the potential for investigations of heat-related illnesses and deaths. A daily high-resolution human thermal index collection (HiTIC-Daily) will be produced and released in our future studies.

2. Lines 168-169: How about the impacts of precipitation on thermal indices? Have the authors considered precipitation as a covariate?

*Response*: Thank you for bringing our attention to precipitation. We did not include precipitation as a covariate because the precipitation data are not normally distributed.

More importantly, they exhibit many zero values in many regions of China (especially in the dry season), which would increase the uncertainty of the spatial prediction.

3. Figures 7 and 8: The results indicate spatial variability of bias in the thermal indices. What factors drive the spatial variability of the bias? Meanwhile, there is also temporal variability in the bias (Figure 8), and what is the drivers of this variability? Are the spatial and temporal variabilities of the bias related to background climates?

*Response*: Thank you for the comment. As shown in Figure 7, the bias exhibits a zonal variation across the space, i.e., positive bias values tend to distribute in northern China and negative values are mainly located in the south. This spatial variability is likely caused by the generally lower temperatures in the north and higher temperatures in the south, and extremely small values in the north may be overestimated while extremely large values in the south may be underestimated to some extent. Such an issue of overestimation of low values and underestimation of high values is a common problem in machine learning prediction (Cho, Yoo, Im, & Cha, 2020; Li, Li, Li, & Liu, 2020; Uddin, Nash, Mahammad Diganta, Rahman, & Olbert, 2022; Wu et al., 2022). The same reason can also explain the temporal variability of the bias (Figure 8). Positive bias values are more likely seen in early periods with lower temperatures, and negative bias values tend to appear in more recent periods with higher temperatures. Although there exist some spatial and temporal variabilities of the bias, these variations are quite small (i.e., ranging from -0.3 °C to +0.3 °C), and the overall estimations in our study are still reliable (see the evaluation results in Section 4.1).

4. One way of evaluating the quality of these products is to evaluate the EOFs of these products. For example, what are the first three EOFs in each product? How do the temporal coefficients change over time across these products? Such spatial-temporal evaluation would be desirable.

*Response*: Thank you very much for this suggestion. We have followed your suggestion and evaluated the first three EOF modes of the 12 thermal indices. The spatial patterns and corresponding temporal coefficients of these EOFs are depicted in Figures R1−R4 (see also Figure S10−S13 in the Supporting Information). As Figure R1 shows, the leading EOF (EOF1) of all 12 thermal indices exhibit highly consistent spatial distribution with higher values in the north and lower values in the south. Their temporal variations are also similar to each other (Figure R2). The second and third EOF modes (EOF2 and EOF3) are also similar among different thermal indices (except EOF3 of NET, Figures R2−R4). These results demonstrate the desirable quality of our products. We have thus included the following discussions in the revised manuscript (Lines 334-340):

"**The dominant modes of these indices are further examined by applying the empirical orthogonal function (EOF) analysis (Figures S10−S13). As Figure S10 shows, the leading EOF (EOF1) of all 12 thermal indices exhibit highly consistent spatial distribution with higher values in the northern region and lower values in the south. Their temporal variations are also similar to each other (Figure S11). The second and third EOF modes (EOF2 and EOF3) are also similar among different thermal indices (except EOF3 of NET, Figures S11−S13). These results demonstrate the desirable quality of our products.**"

[Figure]

**Figure R1. Spatial distributions of the leading empirical orthogonal function (EOF1) of the 12 human thermal indices over the mainland of China. The numbers in square brackets represent the percentage of the variance explained by the corresponding EOF mode.**

[Figure]

**Figure R2. Time series of the principal components (PCs) corresponding to the first three EOF modes of the 12 human thermal indices over the mainland of China during 2003–2020.**

[Figure]

**Figure R3. As Figure R1 but for the second EOF (EOF2).**

[Figure]

**Figure R4. As Figure R1 but for the third EOF (EOF3).**

**#Reviewer 3**

1. Line 157-158, suggest: Comparisons on our products with two existing datasets are in Section 5, data availability is provided in Section 6, ...

*Response*: Revised per your suggestion.

2. The 3rd response to the reviewer2 (spatial variability, temporal variability, background climates), Please consider how to answer the question directly.

*Response*: Thank you very much for your comment. Below please see our updated and direct responses:

As shown in Figure 7, the biases exhibit zonal variations across the space, i.e., positive bias values tend to distribute in northern China and negative values are mainly located in the south. This spatial variability is likely caused by the generally low and high temperatures in the north and south, respectively. The extremely small values in the north may be overestimated while the extremely large values in the south may be underestimated to some extent. The overestimation and underestimation issues are quite common in machine learning (Cho et al., 2020; Li et al., 2020; Uddin et al., 2022; Wu et al., 2022). This can explain the temporal variability of the bias (Figure 8) as well. Positive bias values are more likely to be seen in early periods with lower temperature, and negative bias values tend to appear in more recent periods with higher temperature. Although the biases have spatial and temporal variabilities, these variations are quite small (i.e., ranging from -0.3 °C to +0.3 °C). Overall, the estimations in our study are reliable (see the evaluation results in Section 4.1).

[revised manuscript text omitted]

The copyright of individual parts of the supplement might differ from the article license.

**Figures**

[Figure]

**Figure S1. Monthly prediction accuracies of the 12 human thermal indices over the mainland of China during 2003–2020.**

[Figure]

**Figure S2. Temporal changes of the national average of spring mean human thermal indices over the mainland of China during 2003–2020. The straight line illustrates the linear trend, and the number in the square bracket means the corresponding trend per decade. The asterisk next to the number indicates that the trend is significant at the 0.05 level.**

[Figure]

**Figure S3. As Figure S2 but for summer mean.**

[Figure]

**Figure S4. As Figure S2 but for autumn mean.**

[Figure]

**Figure S5. As Figure S2 but for winter mean.**

[Figure]

**Figure S6. Spatial distributions of the trends (unit: °C/decade) of summer mean human thermal indices over the mainland of China during 2003–2020.**

[Figure]

**Figure S7. As Figure S6 but for winter mean.**

[Figure]

**Figure S8. As Figure S6 but for spring mean.**

[Figure]

**Figure S9. As Figure S6 but for autumn mean.**

[Figure]

**Figure S10. Spatial distributions of the first empirical orthogonal function (EOF1) mode of the 12 human thermal indices over the mainland of China. The number in square brackets represents the percentage of the variance explained by the corresponding EOF mode.**

[Figure]

**Figure S11. Time series of the principal components (PCs) corresponding to the first three EOF modes of the 12 human thermal indices over the mainland of China during 2003–2020.**

[Figure]

**Figure S12. As Figure S10 but for the second EOF (EOF2).**

[Figure]

**Figure S13. As Figure S10 but for the third EOF (EOF3).**

**Tables**

Table S1. $R^2$ of the 12 predicted human thermal indices in the 20 major urban agglomerations (UAs) of the mainland of China during 2003–2020.

| UAs | SAT | $AT_{in}$ | $AT_{out}$ | DI | ET | HI | HMI | MDI | NET | sWBGT | WBT | WCT |
|---|---|---|---|---|---|---|---|---|---|---|---|---|
| Beibu Gulf | 0.9874 | 0.9884 | 0.9881 | 0.9886 | 0.9880 | 0.9851 | 0.9881 | 0.9883 | 0.9862 | 0.9871 | 0.9866 | 0.9872 |
| Beijing-Tianjin-Hebei | 0.9979 | 0.9979 | 0.9978 | 0.9977 | 0.9979 | 0.9974 | 0.9974 | 0.9975 | 0.9964 | 0.9972 | 0.9966 | 0.9974 |
| Central Guizhou | 0.9930 | 0.9931 | 0.9927 | 0.9929 | 0.9928 | 0.9928 | 0.9927 | 0.9926 | 0.9871 | 0.9923 | 0.9912 | 0.9899 |
| Central Henan | 0.9972 | 0.9973 | 0.9973 | 0.9973 | 0.9972 | 0.9967 | 0.9970 | 0.9971 | 0.9946 | 0.9968 | 0.9960 | 0.9957 |
| Central Shanxi | 0.9973 | 0.9974 | 0.9974 | 0.9973 | 0.9973 | 0.9972 | 0.9973 | 0.9971 | 0.9943 | 0.9971 | 0.9965 | 0.9955 |
| Central Yunnan | 0.9892 | 0.9912 | 0.9898 | 0.9912 | 0.9899 | 0.9904 | 0.9913 | 0.9915 | 0.9837 | 0.9913 | 0.9904 | 0.9876 |
| Chengdu-Chongqing | 0.9929 | 0.9936 | 0.9932 | 0.9931 | 0.9929 | 0.9921 | 0.9932 | 0.9926 | 0.9884 | 0.9931 | 0.9910 | 0.9897 |
| Guanzhong | 0.9959 | 0.9959 | 0.9958 | 0.9957 | 0.9958 | 0.9955 | 0.9957 | 0.9952 | 0.9925 | 0.9954 | 0.9943 | 0.9939 |
| Harbin-Changchun | 0.9983 | 0.9984 | 0.9983 | 0.9983 | 0.9983 | 0.9983 | 0.9983 | 0.9983 | 0.9961 | 0.9981 | 0.9980 | 0.9973 |
| Hu-Bao-E-Yu | 0.9977 | 0.9978 | 0.9976 | 0.9977 | 0.9976 | 0.9978 | 0.9977 | 0.9975 | 0.9942 | 0.9975 | 0.9968 | 0.9955 |
| Jiang-Huai | 0.9966 | 0.9970 | 0.9968 | 0.9968 | 0.9968 | 0.9955 | 0.9966 | 0.9967 | 0.9953 | 0.9964 | 0.9960 | 0.9962 |
| Lanzhou-Xining | 0.9964 | 0.9969 | 0.9968 | 0.9969 | 0.9968 | 0.9970 | 0.9969 | 0.9970 | 0.9939 | 0.9969 | 0.9966 | 0.9952 |
| Mid-southern Liaoning | 0.9974 | 0.9973 | 0.9971 | 0.9973 | 0.9975 | 0.9971 | 0.9969 | 0.9971 | 0.9949 | 0.9966 | 0.9966 | 0.9967 |
| Middle Reaches of Yangtze River | 0.9957 | 0.9959 | 0.9956 | 0.9958 | 0.9957 | 0.9943 | 0.9956 | 0.9955 | 0.9933 | 0.9955 | 0.9946 | 0.9946 |
| Ningxia Yellow River | 0.9974 | 0.9978 | 0.9975 | 0.9980 | 0.9978 | 0.9978 | 0.9978 | 0.9978 | 0.9948 | 0.9978 | 0.9972 | 0.9967 |
| North Tianshan Mountain | 0.9955 | 0.9949 | 0.9941 | 0.9942 | 0.9952 | 0.9947 | 0.9943 | 0.9938 | 0.9903 | 0.9938 | 0.9924 | 0.9905 |
| Pearl River Delta | 0.9893 | 0.9901 | 0.9907 | 0.9904 | 0.9899 | 0.9855 | 0.9893 | 0.9902 | 0.9896 | 0.9888 | 0.9889 | 0.9906 |
| Shandong Peninsula | 0.9975 | 0.9976 | 0.9973 | 0.9975 | 0.9976 | 0.9970 | 0.9972 | 0.9973 | 0.9960 | 0.9969 | 0.9965 | 0.9970 |

| | | | | | | | | | | | |
|---|---|---|---|---|---|---|---|---|---|---|---|
| West Coast of Taiwan Strait | 0.9925 | 0.9922 | 0.9928 | 0.9923 | 0.9923 | 0.9897 | 0.9916 | 0.9917 | 0.9904 | 0.9911 | 0.9905 | 0.9920 |
| Yangtze River Delta | 0.9972 | 0.9974 | 0.9971 | 0.9974 | 0.9972 | 0.9960 | 0.9972 | 0.9972 | 0.9961 | 0.9970 | 0.9966 | 0.9969 |

**Table S2.** *RMSE* (°C) of the 12 predicted human thermal indices in the 20 major urban agglomerations (UAs) of the mainland of China during 2003–2020.

| UAs | SAT | AT$_{in}$ | AT$_{out}$ | DI | ET | HI | HMI | MDI | NET | sWBGT | WBT | WCT |
|---|---|---|---|---|---|---|---|---|---|---|---|---|
| Beibu Gulf | 0.626 | 0.713 | 0.786 | 0.587 | 0.382 | 0.908 | 1.031 | 0.671 | 0.757 | 0.672 | 0.634 | 0.725 |
| Beijing-Tianjin-Hebei | 0.513 | 0.548 | 0.618 | 0.515 | 0.315 | 0.640 | 0.781 | 0.607 | 0.699 | 0.494 | 0.612 | 0.649 |
| Central Guizhou | 0.574 | 0.633 | 0.707 | 0.564 | 0.369 | 0.646 | 0.864 | 0.645 | 0.887 | 0.545 | 0.610 | 0.776 |
| Central Henan | 0.492 | 0.532 | 0.595 | 0.475 | 0.309 | 0.622 | 0.747 | 0.557 | 0.748 | 0.471 | 0.567 | 0.704 |
| Central Shanxi | 0.525 | 0.549 | 0.601 | 0.518 | 0.334 | 0.607 | 0.703 | 0.598 | 0.782 | 0.442 | 0.567 | 0.776 |
| Central Yunnan | 0.487 | 0.485 | 0.592 | 0.432 | 0.296 | 0.509 | 0.651 | 0.481 | 0.717 | 0.399 | 0.456 | 0.621 |
| Chengdu-Chongqing | 0.607 | 0.646 | 0.730 | 0.574 | 0.380 | 0.738 | 0.898 | 0.663 | 0.832 | 0.558 | 0.633 | 0.813 |
| Guanzhong | 0.581 | 0.630 | 0.705 | 0.582 | 0.370 | 0.690 | 0.837 | 0.685 | 0.826 | 0.528 | 0.656 | 0.813 |
| Harbin-Changchun | 0.590 | 0.591 | 0.661 | 0.563 | 0.371 | 0.650 | 0.749 | 0.642 | 0.991 | 0.463 | 0.591 | 0.855 |
| Hu-Bao-E-Yu | 0.589 | 0.566 | 0.664 | 0.545 | 0.371 | 0.631 | 0.703 | 0.619 | 0.967 | 0.434 | 0.587 | 0.952 |
| Jiang-Huai | 0.511 | 0.549 | 0.613 | 0.487 | 0.313 | 0.721 | 0.788 | 0.557 | 0.686 | 0.502 | 0.536 | 0.626 |
| Lanzhou-Xining | 0.553 | 0.532 | 0.591 | 0.502 | 0.332 | 0.574 | 0.655 | 0.554 | 0.729 | 0.392 | 0.505 | 0.723 |
| Mid-southern Liaoning | 0.604 | 0.650 | 0.746 | 0.604 | 0.374 | 0.717 | 0.891 | 0.699 | 0.932 | 0.560 | 0.670 | 0.806 |
| Middle Reaches of Yangtze River | 0.547 | 0.606 | 0.686 | 0.526 | 0.345 | 0.773 | 0.857 | 0.610 | 0.765 | 0.534 | 0.584 | 0.702 |
| Ningxia Yellow River | 0.541 | 0.509 | 0.612 | 0.449 | 0.313 | 0.552 | 0.627 | 0.527 | 0.786 | 0.373 | 0.501 | 0.706 |
| North Tianshan Mountain | 0.981 | 1.025 | 1.183 | 1.009 | 0.635 | 1.136 | 1.277 | 1.135 | 1.405 | 0.783 | 1.034 | 1.531 |
| Pearl River Delta | 0.557 | 0.660 | 0.683 | 0.537 | 0.346 | 0.922 | 0.988 | 0.614 | 0.648 | 0.635 | 0.585 | 0.615 |
| Shandong Peninsula | 0.498 | 0.533 | 0.627 | 0.490 | 0.303 | 0.630 | 0.764 | 0.569 | 0.709 | 0.486 | 0.571 | 0.640 |
| West Coast of Taiwan Strait | 0.551 | 0.652 | 0.679 | 0.542 | 0.346 | 0.833 | 0.956 | 0.634 | 0.696 | 0.612 | 0.599 | 0.650 |
| Yangtze River Delta | 0.453 | 0.505 | 0.575 | 0.436 | 0.288 | 0.668 | 0.706 | 0.504 | 0.617 | 0.451 | 0.487 | 0.554 |

**Table S3.** *MAE* (°C) of the 12 predicted human thermal indices in the 20 major urban agglomerations (UAs) of the mainland of China during 2003–2020.

| UAs | SAT | AT$_{in}$ | AT$_{out}$ | DI | ET | HI | HMI | MDI | NET | sWBGT | WBT | WCT |
|---|---|---|---|---|---|---|---|---|---|---|---|---|
| Beibu Gulf | 0.489 | 0.550 | 0.605 | 0.448 | 0.296 | 0.687 | 0.775 | 0.506 | 0.583 | 0.506 | 0.481 | 0.567 |
| Beijing-Tianjin-Hebei | 0.389 | 0.418 | 0.473 | 0.384 | 0.239 | 0.478 | 0.587 | 0.454 | 0.528 | 0.369 | 0.457 | 0.496 |
| Central Guizhou | 0.437 | 0.493 | 0.533 | 0.431 | 0.283 | 0.491 | 0.665 | 0.489 | 0.645 | 0.419 | 0.460 | 0.570 |
| Central Henan | 0.376 | 0.403 | 0.458 | 0.359 | 0.236 | 0.460 | 0.555 | 0.419 | 0.542 | 0.351 | 0.424 | 0.516 |
| Central Shanxi | 0.402 | 0.417 | 0.472 | 0.386 | 0.250 | 0.454 | 0.537 | 0.447 | 0.588 | 0.336 | 0.429 | 0.554 |
| Central Yunnan | 0.377 | 0.380 | 0.459 | 0.332 | 0.230 | 0.395 | 0.505 | 0.371 | 0.545 | 0.308 | 0.349 | 0.475 |
| Chengdu-Chongqing | 0.455 | 0.488 | 0.545 | 0.425 | 0.286 | 0.545 | 0.680 | 0.490 | 0.602 | 0.424 | 0.470 | 0.582 |
| Guanzhong | 0.439 | 0.472 | 0.530 | 0.432 | 0.279 | 0.513 | 0.634 | 0.507 | 0.611 | 0.397 | 0.494 | 0.596 |
| Harbin-Changchun | 0.435 | 0.442 | 0.494 | 0.415 | 0.276 | 0.485 | 0.556 | 0.475 | 0.666 | 0.341 | 0.438 | 0.569 |
| Hu-Bao-E-Yu | 0.458 | 0.434 | 0.517 | 0.407 | 0.281 | 0.485 | 0.538 | 0.466 | 0.689 | 0.332 | 0.448 | 0.629 |
| Jiang-Huai | 0.387 | 0.421 | 0.480 | 0.370 | 0.240 | 0.528 | 0.598 | 0.424 | 0.536 | 0.379 | 0.409 | 0.488 |
| Lanzhou-Xining | 0.432 | 0.408 | 0.462 | 0.386 | 0.261 | 0.446 | 0.503 | 0.429 | 0.571 | 0.302 | 0.393 | 0.553 |
| Mid-southern Liaoning | 0.465 | 0.495 | 0.577 | 0.458 | 0.286 | 0.541 | 0.671 | 0.530 | 0.692 | 0.423 | 0.512 | 0.610 |
| Middle Reaches of Yangtze River | 0.418 | 0.464 | 0.521 | 0.397 | 0.263 | 0.575 | 0.651 | 0.459 | 0.567 | 0.406 | 0.440 | 0.529 |
| Ningxia Yellow River | 0.422 | 0.400 | 0.486 | 0.343 | 0.242 | 0.423 | 0.483 | 0.409 | 0.592 | 0.291 | 0.372 | 0.535 |
| North Tianshan Mountain | 0.715 | 0.772 | 0.861 | 0.743 | 0.460 | 0.838 | 0.988 | 0.843 | 0.963 | 0.602 | 0.783 | 0.977 |
| Pearl River Delta | 0.428 | 0.492 | 0.522 | 0.399 | 0.261 | 0.663 | 0.715 | 0.455 | 0.495 | 0.459 | 0.427 | 0.470 |
| Shandong Peninsula | 0.379 | 0.408 | 0.477 | 0.372 | 0.234 | 0.475 | 0.577 | 0.432 | 0.522 | 0.364 | 0.433 | 0.482 |
| West Coast of Taiwan Strait | 0.427 | 0.509 | 0.530 | 0.425 | 0.269 | 0.623 | 0.748 | 0.492 | 0.530 | 0.474 | 0.464 | 0.501 |
| Yangtze River Delta | 0.346 | 0.378 | 0.438 | 0.324 | 0.217 | 0.476 | 0.525 | 0.373 | 0.470 | 0.331 | 0.360 | 0.425 |

**Table S4.** *Bias* (°C) of the 12 predicted human thermal indices in the 20 major urban agglomerations of the mainland of China during 2003–2020.

| UAs | SAT | $AT_{in}$ | $AT_{out}$ | DI | ET | HI | HMI | MDI | NET | sWBGT | WBT | WCT |
|---|---|---|---|---|---|---|---|---|---|---|---|---|
| Beibu Gulf | -0.072 | -0.089 | -0.121 | -0.079 | -0.037 | -0.094 | -0.147 | -0.082 | -0.096 | -0.098 | -0.108 | -0.106 |
| Beijing-Tianjin-Hebei | 0.052 | 0.073 | 0.073 | 0.069 | 0.034 | 0.080 | 0.119 | 0.086 | 0.024 | 0.096 | 0.059 | 0.079 |
| Central Guizhou | 0.008 | -0.004 | -0.020 | -0.018 | 0.005 | -0.008 | -0.023 | -0.029 | -0.008 | -0.037 | 0.010 | -0.024 |
| Central Henan | 0.009 | 0.023 | -0.009 | 0.026 | 0.005 | 0.024 | 0.051 | 0.035 | -0.015 | 0.040 | -0.020 | 0.037 |
| Central Shanxi | 0.037 | 0.027 | 0.031 | 0.031 | 0.016 | 0.036 | 0.047 | 0.034 | 0.041 | 0.037 | 0.038 | 0.028 |
| Central Yunnan | -0.008 | -0.019 | -0.057 | -0.018 | -0.007 | -0.011 | -0.041 | -0.023 | -0.039 | -0.029 | -0.055 | -0.020 |
| Chengdu-Chongqing | -0.063 | -0.053 | -0.075 | -0.054 | -0.029 | -0.056 | -0.082 | -0.071 | -0.064 | -0.063 | -0.064 | -0.043 |
| Guanzhong | 0.002 | 0.014 | -0.009 | 0.012 | 0.013 | 0.015 | 0.022 | 0.017 | 0.001 | 0.013 | 0.022 | 0.014 |
| Harbin-Changchun | 0.008 | 0.011 | 0.014 | 0.009 | 0.001 | 0.010 | 0.028 | 0.011 | 0.007 | 0.012 | 0.006 | 0.017 |
| Hu-Bao-E-Yu | 0.027 | 0.007 | 0.051 | 0.003 | 0.003 | 0.008 | 0.022 | 0.011 | 0.021 | 0.012 | 0.003 | 0.014 |
| Jiang-Huai | 0.020 | -0.022 | -0.016 | -0.035 | -0.015 | -0.020 | -0.030 | -0.029 | -0.012 | -0.035 | -0.014 | -0.023 |
| Lanzhou-Xining | 0.018 | 0.035 | 0.040 | 0.043 | 0.020 | 0.040 | 0.053 | 0.038 | 0.069 | 0.044 | 0.048 | 0.031 |
| Mid-southern Liaoning | 0.078 | 0.086 | 0.104 | 0.079 | 0.043 | 0.090 | 0.123 | 0.091 | 0.111 | 0.085 | 0.103 | 0.084 |
| Middle Reaches of Yangtze River | -0.019 | -0.028 | -0.030 | -0.023 | -0.016 | -0.028 | -0.030 | -0.030 | -0.022 | -0.029 | -0.027 | -0.017 |
| Ningxia Yellow River | 0.028 | -0.017 | 0.006 | -0.001 | -0.001 | -0.007 | -0.003 | 0.000 | 0.003 | 0.025 | 0.056 | -0.006 |
| North Tianshan Mountain | -0.108 | 0.015 | -0.004 | -0.003 | -0.017 | -0.029 | 0.064 | -0.014 | -0.096 | 0.016 | -0.145 | 0.048 |
| Pearl River Delta | -0.089 | -0.105 | -0.083 | -0.085 | -0.061 | -0.154 | -0.160 | -0.099 | -0.065 | -0.102 | -0.069 | -0.095 |
| Shandong Peninsula | 0.039 | 0.042 | 0.072 | 0.033 | 0.019 | 0.051 | 0.058 | 0.039 | 0.070 | 0.046 | 0.048 | 0.038 |
| West Coast of Taiwan Strait | -0.029 | -0.058 | -0.040 | -0.047 | -0.023 | -0.056 | -0.091 | -0.056 | -0.039 | -0.057 | -0.036 | -0.052 |
| Yangtze River Delta | 0.000 | -0.020 | -0.021 | -0.021 | -0.011 | -0.024 | -0.022 | -0.023 | -0.019 | -0.018 | -0.024 | -0.020 |

---

## Author Response (AR2)

**Responses to Review Comments**

Dear Editor,

Thank you and all the reviewers for their careful review and valuable comments. We have carefully addressed all of the comments and revised our manuscript accordingly. Below please find our item-by-item responses to all comments raised by two reviewers and the topic editor. Please note that the review comments are in black, our responses are highlighted in blue, the extracts from the manuscript are in red, and **the new texts that have been added or changed** are in bold. Thank you very much for your time.

Best regards,

Ming Luo and Yongquan Zhao, on behalf of all authors

**Comments by Reviewer #1**

Major comments.

The dataset is very import to study the climate change during this period over this region. The high spatial resolution and the monthly temporal resolution are basically reasonable, based on the previous datasets. The use of homogenous meteorological observation data is the right way to evaluate. This research is novel and belongs to the application of big data for large volume of Satellite data used, and machine learning algorithm. The authors are from 8 universities/colleges. The results supply important reference for other research fields like the field of geoscience, biology, sociology, and medicine and engineering.

*Response*: Many thanks for your appreciation of our work and the constructive comments. We have carefully revised our manuscript according to your comments and gave the following responses for your further review.

1) Line 415, harm, it is suggested to change the word. All 11 indices are calculated based on SAT, so the accuracy of this dataset is up to the accuracy of the dataset (Zhang 2022b) and the algorithm. SAT has important influence to the other 11 indices. How to understand these?

*Response*: Thank you for your comment. We have changed "harm" to "influence". The 11 human thermal indices were calculated from SAT at a daily scale (see Table 2 in the main text) for each station. Then, the daily thermal indices were monthly averaged and used for training together with the other predictors (e.g., LST, land cover, topography, and temporal variation) to estimate the spatial distribution of the 11 monthly indices at the 1 km grids. Ideally, the spatial prediction should be conducted based on monthly or even daily SAT. However, the SAT observations with full spatial coverage are not available, so we used monthly LST as an alternative to predict the spatial distribution of the monthly human thermal indices. In this case, the accuracy of monthly LST may influence the accuracy of our HiTIC-monthly dataset. We thereby performed multiple

accuracy assessments at various spatial (e.g., individual stations, the mainland of China, and its urban agglomerations) and temporal scales (e.g., monthly, yearly, and overall). The assessment results indicate that our dataset shows a desirable performance and exhibits good agreement with the observations in both spatial and temporal dimensions (see Section 4.1 of the main text), demonstrating the broad applicability of our dataset.

2) (Zhang 2022b) SAT is LST, but not Tair (1.5 meters above the surface). Why use LST but not Tair in Table 1 to compute the other 11 indices?

*Response*: As shown in our response to your comment #1, we first computed daily thermal indices observed at the weather stations based on daily SAT and other meteorological observations (see Table 2), and then aggregated them to monthly values. These spatially-discrete (station-based) monthly indices were then used for predicting spatially-continuous human thermal indices on 1 km grids, which adopted seamless monthly LST (seamless SAT is not available) and other related variables as predictors. Previous studies have shown that LST is highly correlated with SAT and can be used as a proxy to predict SAT and SAT-related human thermal indices (Shamir & Georgakakos, 2014; W. Zhu, Lü, & Jia, 2013; X. Zhu, Zhang, Xu, Sun, & Hu, 2019). Our results also demonstrate that using LST can generate a desirable accuracy in predicting monthly human thermal indices.

3) Line 157-158 "Section 6 compares our products with two existing datasets, and the main findings of this paper are summarized in Section 7". Where is section 6 and section 7? They are data availability and conclusions.

*Response*: Thank you for pointing out this issue. We have revised the description as (Lines 157–159): "**Comparisons on our products with two existing datasets are in Section 5, data availability is provided in Section 6. The main findings of this paper are summarized in Section 7.**"

4) Why not keep Figures S1-S9, Tables S1-S5 as formal ones? Please consider again which to keep in the manuscript.

*Response*: Thanks a lot for your advice. We have moved Tables S1–S2 to the main text in the current revision, and we decided to keep Figures S1–S9 and Tables S3–S5 in the supplementary material because there are already 14 figures and 4 tables in the main text. Including more figures and tables may not meet the requirement of the journal of ESSD and influence the readability of our manuscript.

5) How to get monthly data from MODIS daily mid-daytime 13:30 and mid-nighttime 01:30 LST? The monthly mean is different considering of the diurnal variation and satellite observations from ascending orbits and descending orbits.

*Response*: Thank you for the comment. Our monthly LST values were calculated by averaging daily LST in the corresponding month in the calendar years of 2003−2020. The daily mean LST values were obtained by averaging four MODIS observations in a day, which correspond to mid-daytime and mid-nighttime observations from ascending and descending orbits. In the current revised version, we have added the following explanations (Lines 178–183): "**We used monthly LST as one of the inputs to predict the spatial distribution of 12 thermal indices. Monthly LST values were calculated by averaging daily LST, which was obtained by averaging four observations in a day, including mid-daytime and mid-nighttime observations from ascending and descending orbits of MOD11A1 (Terra) and MYD11A1 (Aqua). More details about the LST data are described in Zhang et al. (2022b).**"

6) Line 190-191, how to compute and use the covariates?

*Response*: Thank you for the question. The covariates directly come from the data providers or producers, and these information is listed in Table 1. All these covariates

were pre-processed to have the same spatial extent, projection, and spatial resolution for predicting human thermal indices with full spatial coverage at the 1 km resolution.

7) HiTIC? What is i? Is it HTI (human thermal index)? No "I" after "human" and before "thermal".

 *Response*: "HiTIC" stands for **Hi**gh spatial resolution **T**hermal **I**ndex **C**ollection. We have changed the title to "**HiTIC-Monthly: A Monthly High Spatial Resolution (1 km) Human Thermal Index Collection over China during 2003−2020**" to avoid misunderstanding.

Minor comments.

1) Add abbr. of LGBM when it is first used. Light Gradient Boosting Machine. Check others, please.

*Response*: Thank you for your suggestion. The full name of LGBM has been added (Lines 38–39): "In this collection, 12 commonly-used thermal indices were generated by **the Light Gradient Boosting Machine (LGBM)** learning algorithm from multi-source gridded data."

2) Author contribution. It seems only H.Z is responsible for analysis and data processing, others are all involved in writing. For computations, are there anyone else in the author list?

*Response*: The first author H.Z. and two corresponding authors of M.L. and Y.Z. are responsible for data analysis and computation. We also wish to clarify that this work involves a large number of tasks (including but not limited to the selection of predictors, the accuracy assessment, the discussion and interpretation of the results, and the comparison with other datasets), and all other authors in the list contributed to discussion, investigation, and writing.

3) Line 439. The unit of the dataset is degree or 0.01 degree? Is 0.01 the scale factor?

*Response*: Yes, the unit of the dataset is 0.01 degree Celsius (°C). The values are stored in integer type (Int16) for saving storage and need to be multiplied by 100 when in use. The following updates have been made in the main text (Lines 472–474): "The unit of the dataset is 0.01 degree Celsius (°C)**, and the values are stored in an integer type (Int16) for saving storage space, and need to be divided by 100 to get the values in degree Celsius when in use.**"

4) Please revise the followings.

● Line 247-248, as the figure displays

*Response*: Revised.

● Line 384, Figures S8j&m. "Figure S8", not figures, and add a blank before j.

*Response*: Revised.

It is suggested to delete equations (1-2) and (3-6), add the reference.

*Response*: Thanks for the suggestion. With respect, we believe that these equations are helpful for readers to understand the details more easily, and thus can be kept in the main text. As suggested, we have added the relevant references in the current revision:

Lines 205–206: "$E_a$ is derived from T and RH rather than directly observed at meteorological stations (Eqs. 1~2; **Bolton (1980)**)."

Lines 238–239: "Four statistic metrics, namely, determination coefficient ($R^2$), Mean Absolute Error (*MAE*), *RMSE*, and *Bias* **(Rice, 2006)**."

● If necessary, please keep the same description of the time period for there are many kinds in the manuscript, like 2003~2020, from 2003 to 2020, during the year

2003~2020, (2003 to 2020), during 2003~2020, from January 2003 to December 2020, 2003-2020.

*Response*: Thanks for your suggestion. We have changed all descriptions of the time period to "**during 2003−2020**".

- Line 755. Delete the last sentence. Line 759, change : to . after 2003~2020.

*Response*: Revised.

- Add longitude and latitude (or the numerical scope signs) in Fig 1, 4-7, 9-12, 14.

*Response*: Added.

Figure 8. Prediction accuracies of 12 human thermal indices… Add 12. In individual years, change the description. Time series? Add the description of (a) (b) (c).

*Response*: Added.

- Figure 10. Spatial distributions of 12 human thermal indices... Add 12.

*Response*: Added.

- Line 780. Figure 11. National average, are you sure to write like this? Check it in other places in the manuscript. Just use average is fine. Delete "straight". Line 789, Figure 13, national, delete this. Add 12, 12 human thermal indices.

*Response*: Thank you. We have revised these sentences accordingly. For example, we have revised the caption of Figure 11 as (Lines 843–844): "Temporal changes of **the 12 annually-averaged human thermal indices over the mainland of China during 2003−2020**."

- Line 785, Figure 12, add 12. "the trends of annual mean", please consider how to express better. Inter-annual variation?

*Response*: Thank you for the comment. The sentence has been rewritten as (Lines 849–850): "Spatial distributions of the **linear** trends (unit: °C per decade) in **the 12**

**annually-averaged human thermal indices** over the mainland of China **during 2003−2020**."

- Line 795, Figure 14. HITIC, is it "i"? in mainland China, is it over mainland of China? Please check the description of "mainland China" in the manuscript. In July 2018, move to the end of four major UAs. Delete i.e., change to :.

*Response*: Thank you for the comment. We have revised these sentences as follows (Lines 858-861):

"Comparison of the spatial patterns among HDI_0p25_1970_2018 (HDI), HiTiSEA, and **HiTIC-Monthly** for $AT_{in}$ **over the mainland of China and its four largest UAs** in July 2018: Beijing-Tianjin-Hebei (BTH), Yangtze River Delta (YRD), middle Yangtze River Valley (mYRV) and Pearl River Delta (PRD). Colored circles indicate the observed $AT_{in}$ values at individual meteorological stations."

**Comments by Reviewer #2**

The authors have produced a high-resolution (1 km×1 km) thermal index collection at a monthly scale (HiTIC-Monthly) in China during 2003 to 2020, with 12 widely used human thermal indices. The authors have created a high-resolution products for quantifying thermal index in China, which is valuable to the scientific community. I have some comments to be addressed by the authors.

Response: Thank you very much for your helpful comments and suggestions. We have carefully revised our manuscript based on your comments. Below please find our item-by-item responses.

1. The biggest concern is the temporal resolution. Why do the authors choose monthly resolution, rather than daily? Daily products would be extremely useful to characterize extreme events, which are of societal importance.

*Response*: Thank you very much for your comment. We agree with you that daily products could be useful to characterize extreme events. However, a daily thermal indices dataset with full spatial coverage requires daily covariates that have full spatial coverage as well, and these data are not available. We produced a long-term monthly thermal indices dataset with high accuracies. The monthly dataset could be useful for the studies of climate change and urban climate and environment across China and has the potential for investigations of heat-related illnesses and deaths. A daily high-resolution human thermal index collection (HiTIC-Daily) will be produced and released in our future studies.

2. Lines 168-169: How about the impacts of precipitation on thermal indices? Have the authors considered precipitation as a covariate?

*Response*: Thank you for bringing our attention to precipitation. We did not include precipitation as a covariate because the precipitation data are not normally distributed.

More importantly, they exhibit many zero values in many regions of China (especially in the dry season), which would increase the uncertainty of the spatial prediction.

3. Figures 7 and 8: The results indicate spatial variability of bias in the thermal indices. What factors drive the spatial variability of the bias? Meanwhile, there is also temporal variability in the bias (Figure 8), and what is the drivers of this variability? Are the spatial and temporal variabilities of the bias related to background climates?

*Response*: Thank you for the comment. Yes, the spatial and temporal variabilities of the bias are related to background climates. As shown in Figure 7, the biases exhibit zonal variations across the space, i.e., positive bias values tend to distribute in northern China and negative values are mainly located in the south. The spatial variability is likely caused by the generally lower temperatures in the north and higher temperatures in the south. The extremely small values in the north and the extremely large values in the south may be overestimated and underestimated to some extent, respectively. The overestimation and underestimation issues caused by limited training samples of extreme values are quite common in machine learning (Cho, Yoo, Im, & Cha, 2020; Li, Li, Li, & Liu, 2020; Uddin, Nash, Mahammad Diganta, Rahman, & Olbert, 2022; Wu et al., 2022), due to limited samples of extreme low and high values compared to the rest of the samples. The temporal variability of the bias is relative to climate warming. Under climatic warming, the lower temperatures appear in early periods (e.g., 2003–2005) while relatively higher temperatures occur in more recent periods (e.g., 2016–2019). The overestimation of lower temperature in early periods and the underestimation of higher temperature in recent periods result in the temporal variability of the bias (Figure 8). It should be noted that, although the bias has spatial and temporal variabilities, these variations are quite small (i.e., ranging from -0.3 °C to +0.3 °C). Overall, the estimations in our study are reliable (see the evaluation results in Section 4.1).

In this revision, we have added the following discussions:

Lines 282–290: "**Positive *Bias* values tend to distribute in northern China while negative values are mainly located in the south. This spatial variability is likely caused by the generally lower temperatures in the north and higher temperatures in the south. In particular, the extremely small values in the north and the extremely large values in the south may be overestimated and underestimated to some extent, respectively, due to limited samples of extremely small and large values (compared with the rest of the samples) when training the machine learning model. The overestimation and underestimation issues caused by limited training samples of extreme values are quite common in machine learning (Cho et al., 2020; Li et al., 2020; Uddin et al., 2022; Wu et al., 2022).** "

Lines 297–304: "*Biases* vary between -0.04 °C and 0.04 °C across all years. **This temporal variability of the *Bias* is related to the yearly climate variations, and is characterized by a marginal overestimation of lower temperatures that mainly appeared in early periods (e.g., 2003–2005) and the underestimation of higher temperatures mostly in recent periods (e.g., 2016–2019). Under climatic warming over the past decades, the lower temperatures tended to appear in early periods while relatively higher temperatures more likely occurred in more recent periods. Extremely small values of temperature in earlier periods and the large values in the later periods may be slightly overestimated (i.e., with positive *Bias* values) and underestimated (i.e., with negative *Bias* values), respectively, thereby characterizing the temporal variations of the *Bias*.**".

4. One way of evaluating the quality of these products is to evaluate the EOFs of these products. For example, what are the first three EOFs in each product? How do the temporal coefficients change over time across these products? Such spatial-temporal evaluation would be desirable.

*Response*: Thank you very much for this suggestion. We have followed your suggestion and evaluated the first three EOF modes of the 12 thermal indices. The spatial patterns

and corresponding temporal coefficients of these EOFs are depicted in Figures R1−R4 (see also Figure S10−S13 in the Supporting Information). As Figure R1 shows, the leading EOF (EOF1) of all 12 thermal indices exhibit highly consistent spatial distribution with higher values in the north and lower values in the south. Their temporal variations are also similar to each other (Figure R2). The second and third EOF modes (EOF2 and EOF3) are also similar among different thermal indices (except EOF3 of NET, Figures R2−R4). These results demonstrate the desirable quality of our products. We have thus included the following discussions in the revised manuscript (Lines 350-356):

**"The dominant modes of these indices are further examined by applying the empirical orthogonal function (EOF) analysis (Figures S10−S13). As Figure S10 shows, the leading EOF (EOF1) of all 12 thermal indices exhibit highly consistent spatial distribution with higher values in the northern region and lower values in the south. Their temporal variations are also similar to each other (Figure S11). The second and third EOF modes (EOF2 and EOF3) are also similar among different thermal indices (except EOF3 of NET, Figures S11−S13). These results demonstrate the desirable quality of our products."**

[Figure]

**Figure R1. Spatial distributions of the leading empirical orthogonal function (EOF1) of the 12 human thermal indices over the mainland of China. The numbers in square brackets represent the percentage of the variance explained by the corresponding EOF mode.**

[Figure]

**Figure R2. Time series of the principal components (PCs) corresponding to the first three EOF modes of the 12 human thermal indices over the mainland of China during 2003–2020.**

[Figure]

**Figure R3. As Figure R1 but for the second EOF (EOF2).**

[Figure]

**Figure R4. As Figure R1 but for the third EOF (EOF3).**

**Second-round Comments by Reviewer #1**

1. Line 157-158, suggest: Comparisons on our products with two existing datasets are in Section 5, data availability is provided in Section 6, ...

*Response*: Revised per your suggestion.

2. The 3rd response to the reviewer2 (spatial variability, temporal variability, background climates), Please consider how to answer the question directly.

*Response*: Thank you very much for your comment. Below please see our updated and direct responses (please note that our response to reviewer #2 has also been updated accordingly in the current version):

Yes, the spatial and temporal variabilities of the bias are related to background climates. As shown in Figure 7, the biases exhibit zonal variations across the space, i.e., positive bias values tend to distribute in northern China and negative values are mainly located in the south. The spatial variability is likely caused by the generally lower temperatures in the north and higher temperatures in the south. The extremely small values in the north and the extremely large values in the south may be overestimated and underestimated to some extent, respectively. The overestimation and underestimation issues caused by limited training samples of extreme values are quite common in machine learning (Cho et al., 2020; Li et al., 2020; Uddin et al., 2022; Wu et al., 2022), due to limited samples of extreme low and high values compared to the rest of the samples. The temporal variability of the bias is relative to climate warming. Under climatic warming, the lower temperatures appear in early periods (e.g., 2003–2005) while relatively higher temperatures occur in more recent periods (e.g., 2016–2019). The overestimation of lower temperature in early periods and the underestimation of higher temperature in recent periods result in the temporal variability of the bias (Figure 8). It should be noted that, although the bias has spatial and temporal variabilities, these variations are quite small (i.e., ranging from -0.3 °C to

+0.3 °C). Overall, the estimations in our study are reliable (see the evaluation results in Section 4.1).

In this revision, we have added the following discussions:

Lines 282–290: "**Positive *Bias* values tend to distribute in northern China while negative values are mainly located in the south. This spatial variability is likely caused by the generally lower temperatures in the north and higher temperatures in the south. In particular, the extremely small values in the north and the extremely large values in the south may be overestimated and underestimated to some extent, respectively, due to limited samples of extremely small and large values (compared with the rest of the samples) when training the machine learning model. The overestimation and underestimation issues caused by limited training samples of extreme values are quite common in machine learning (Cho et al., 2020; Li et al., 2020; Uddin et al., 2022; Wu et al., 2022).** "

Lines 297–304: "*Biases* vary between -0.04 °C and 0.04 °C across all years. **This temporal variability of Bias is related to the yearly climate variations, and is characterized by a marginal overestimation of lower temperatures that mainly appeared in early periods (e.g., 2003–2005) and the underestimation of higher temperatures mostly in recent periods (e.g., 2016–2019). Under climatic warming over the past decades, the lower temperatures tended to appear in early periods while relatively higher temperatures more likely occurred in more recent periods. Extremely small values of temperature in earlier periods and the large values in the later periods may be slightly overestimated (i.e., with positive *Bias* values) and underestimated (i.e., with negative *Bias* values), respectively, thereby characterizing the temporal variations of the *Bias*.**"

**Comments by Topic Editor**

There is still a minor comment from one of the reviewers. In order to save time for the publication of the MS, I forward his opinion to you here. Please consider it.

The 3rd response to the reviewer 2 is still not directly answered. The question from the reviewer 2 is good, but the authors almost use the same answer on Oct 1st. What factors drive the spatial variability of the bias? What are the drivers of the temporal variability? Are the spatial and temporal variabilities of the bias related to background climates? If the answers are in the listed references, how about answer the questions directly and add some discussion or explanations in the manuscript?

*Response*: Thank you very much for your comment. Below please see our updated and direct responses. Please be reminded that, in the current version we have also updated our responses to the corresponding comment raised by reviewers #2 and #3.

Yes, the spatial and temporal variabilities of the bias are related to background climates. As shown in Figure 7, the biases exhibit zonal variations across the space, i.e., positive bias values tend to distribute in northern China and negative values are mainly located in the south. The spatial variability is likely caused by the generally lower temperatures in the north and higher temperatures in the south. The extremely small values in the north and the extremely large values in the south may be overestimated and underestimated to some extent, respectively. The overestimation and underestimation issues caused by limited training samples of extreme values are quite common in machine learning (Cho et al., 2020; Li et al., 2020; Uddin et al., 2022; Wu et al., 2022), due to limited samples of extreme low and high values compared to the rest of the samples. The temporal variability of the bias is relative to climate warming. Under climatic warming, the lower temperatures appear in early periods (e.g., 2003–2005) while relatively higher temperatures occur in more recent periods (e.g., 2016–2019). The overestimation of lower temperature in early periods and the underestimation of higher temperature in recent periods result in the temporal variability of the bias (Figure 8). It should be noted that, although the bias has spatial and temporal variabilities, these

variations are quite small (i.e., ranging from -0.3 °C to +0.3 °C). Overall, the estimations in our study are reliable (see the evaluation results in Section 4.1).

In this revision, we have added the following discussions:

Lines 282–290: "**Positive *Bias* values tend to distribute in northern China while negative values are mainly located in the south. This spatial variability is likely caused by the generally lower temperatures in the north and higher temperatures in the south. In particular, the extremely small values in the north and the extremely large values in the south may be overestimated and underestimated to some extent, respectively, due to limited samples of extremely small and large values (compared with the rest of the samples) when training the machine learning model. The overestimation and underestimation issues caused by limited training samples of extreme values are quite common in machine learning (Cho et al., 2020; Li et al., 2020; Uddin et al., 2022; Wu et al., 2022).** "

Lines 297–304: "*Biases* vary between -0.04 °C and 0.04 °C across all years. **This temporal variability of *Bias* is related to the yearly climate variations, and is characterized by a marginal overestimation of lower temperatures that mainly appeared in early periods (e.g., 2003–2005) and the underestimation of higher temperatures mostly in recent periods (e.g., 2016–2019). Under climatic warming over the past decades, the lower temperatures tended to appear in early periods while relatively higher temperatures more likely occurred in more recent periods. Extremely small values of temperature in earlier periods and the large values in the later periods may be slightly overestimated (i.e., with positive *Bias* values) and underestimated (i.e., with negative *Bias* values), respectively, thereby characterizing the temporal variations of the *Bias*.**"

**References**

Cho, D., Yoo, C., Im, J., & Cha, D. H. (2020). Comparative Assessment of Various Machine Learning-Based Bias Correction Methods for Numerical Weather Prediction Model Forecasts of Extreme Air Temperatures in Urban Areas. *Earth and Space Science, 7*(4). doi:https://doi.org/10.1029/2019ea000740

Li, Y., Li, M., Li, C., & Liu, Z. (2020). Forest aboveground biomass estimation using Landsat 8 and Sentinel-1A data with machine learning algorithms. *Sci Rep, 10*(1), 9952. doi:https://doi.org/10.1038/s41598-020-67024-3

Shamir, E., & Georgakakos, K. P. (2014). MODIS Land Surface Temperature as an index of surface air temperature for operational snowpack estimation. *Remote Sensing of Environment, 152*, 83-98. doi:https://doi.org/10.1016/j.rse.2014.06.001

Uddin, M. G., Nash, S., Mahammad Diganta, M. T., Rahman, A., & Olbert, A. I. (2022). Robust machine learning algorithms for predicting coastal water quality index. *J Environ Manage, 321*, 115923. doi:https://doi.org/10.1016/j.jenvman.2022.115923

Wu, J., Fang, H., Qin, W., Wang, L., Song, Y., Su, X., & Zhang, Y. (2022). Constructing High-Resolution (10 km) Daily Diffuse Solar Radiation Dataset across China during 1982–2020 through Ensemble Model. *Remote Sensing, 14*(15). doi:https://doi.org/10.3390/rs14153695

Zhu, W., Lŭ, A., & Jia, S. (2013). Estimation of daily maximum and minimum air temperature using MODIS land surface temperature products. *Remote Sensing of Environment, 130*, 62-73. doi:https://doi.org/10.1016/j.rse.2012.10.034

Zhu, X., Zhang, Q., Xu, C. Y., Sun, P., & Hu, P. (2019). Reconstruction of high spatial resolution surface air temperature data across China: A new geo-intelligent multisource data-based machine learning technique. *Sci Total Environ, 665*, 300-313. doi:https://doi.org/10.1016/j.scitotenv.2019.02.077